# CyberThreat-Eval: Can Large Language Models Automate Real-World Threat Research?

**Xiangsen Chen**[13]* **Xuan Feng**[1], **Shuo Chen**[1], **Matthieu Maitre**[2], **Sudipto Rakshit**[2],
**Diana Duvieilh**[2], **Ashley Picone**[2], **Nan Tang**[3,4]
[1]*Microsoft Research*
[2]*Microsoft*
[3]*Hong Kong University of Science and Techonology (Guangzhou)*
[4]*Hong Kong University of Science and Techonology*

**Reviewed on OpenReview:** https://openreview.net/forum?id=tiFtZHwr7O

## Abstract

Analyzing Open Source Intelligence (OSINT) from large volumes of data is critical for drafting and publishing comprehensive CTI reports. This process usually follows a three-stage workflow—triage, deep search and TI drafting. While Large Language Models (LLMs) offer a promising route toward automation, existing benchmarks still have limitations. These benchmarks often consist of tasks that do not reflect real-world analyst workflows. For example, human analysts rarely receive tasks in the form of multiple-choice questions. Also, existing benchmarks often rely on model-centric metrics that emphasize lexical overlap rather than actionable, detailed insights essential for security analysts. Moreover, they typically fail to cover the complete three-stage workflow. To address these issues, we introduce CyberThreat-Eval, which is collected from the daily CTI workflow of a world-leading company. This expert-annotated benchmark assesses LLMs on practical tasks across all three stages as mentioned above. It utilizes analyst-centric metrics that measure factual accuracy, content quality, and operational costs. Our evaluation using this benchmark reveals important insights into the limitations of current LLMs. For example, LLMs often lack the nuanced expertise required to handle complex details and struggle to distinguish between correct and incorrect information. To address these challenges, the CTI workflow incorporates both external ground-truth databases and human expert knowledge. TRA allows human experts to iteratively provide feedback for continuous improvement. The code of CyberThreat-Eval benchmark is available at https://github.com/secintelligence/CyberThreat-Eval.

## 1 Introduction

Open Source Intelligence (OSINT) reports play a critical role in identifying and researching emerging cybersecurity threats and supporting Cyber Threat Intelligence (CTI) tasks Zhou et al. (2022); Ainslie et al. (2023). Analysts rely on substantial volumes of OSINT data, including threat reports, vulnerability disclosures, and information from social networks Van Puyvelde & Tabárez Rienzi (2025), to draft and publish comprehensive reports. In our production environment, this process unfolds in three stages (Figure 1): triage, in which articles are prioritized; deep search, where supplementary evidence is gathered; and TI drafting, where narratives containing elements such as Indicators of Compromise (IoCs) Microsoft Corporation (2025a) and ATT&CK techniques are compiled into the final report. According to a survey conducted with analysts from our team, analyzing an incident described in a given article typically requires several hours.

LLMs OpenAI (2025); Hurst et al. (2024); Achiam et al. (2023); Liu et al. (2024a); Touvron et al. (2023) offer significant potential to automate these traditionally manual analyses. Recent research demonstrates

---
*Work done during an internship at Microsoft Research Asia.

LLM capability in tasks such as IoC extraction Chen et al. (2025), cross-source correlation Wu et al. (2024); Perrina et al. (2023), and malware or threat actor identification Siracusano et al. (2023). To quantify progress more systematically, the community has also released a series of dedicated CTI benchmarks, including CTIBench for multi-choice reasoning and ATT&CK attribution Alam et al. (2024), CTISum for attack-process summarisation Peng et al. (2024), SECURE for ATT&CK and CVE reasoning Bhusal et al. (2024), CyberMetric for general cybersecurity knowledge evaluation Tihanyi et al. (2024), and CyberBench for report summarisation and threat-category classification Liu et al. (2024b).

**New problem.** Although current benchmarks cover various OSINT and CTI subtasks using diverse automatic metrics, several research gaps remain. **(1) Unrealistic task formats.** Many previous evaluations employ unrealistic tasks, such as multiple-choice or fact-recall questions Alam et al. (2024); Ji et al. (2024); Tihanyi et al. (2024); Liu et al. (2024b). Security analysts almost never attribute breaches to threat actors by selecting from options '*A/B/C/D*'. In addition, Q&A tasks reward rote memory (e.g., What does CVE-2021-26855 refer to?) Such formats fail to evaluate critical analyst skills, including linking evidence to threat actors, assessing business risks, and producing coherent incident summaries. This highlights the need to design more realistic tasks that align with the actual workflows and decision-making processes of security analysts. **(2) The evaluation metrics for some tasks are *model-centric*, not *analyst-centric*.** For example, ROUGE Lin (2004) and BERTScore Zhang et al. (2019) are the main metrics for summary tasks like CTISum Peng et al. (2024). However, in our pilot test a sparse summary outscored a detailed one despite analyst preference for the latter (full example in Appendix E.1). Although human A-B evaluation Peng et al. (2024) is used to assess summaries, it covers only a small portion of the benchmark, lacks clear scoring criteria, and is difficult to scale consistently. Therefore, a new benchmark with more *analyst-centric* metrics should be constructed. **(3) The current benchmarks miss evaluations in key workflow stages.** As Figure 1 shows, the typical OSINT analysis workflow consists of the three stages: triage, deep search, and TI drafting. However, as Table 2 shows, no public benchmark covers all three steps that analysts take in the end-to-end workflow. Current benchmarks mostly focus on single tasks like testing the LLMs to read one threat incident and answer simple questions like *"What is the threat actor of XXX incident?"* Therefore, existing benchmarks typically cover only certain tasks within the three workflow stages mentioned above, while a benchmark that spans the entire end-to-end workflow is still lacking. With the limitations mentioned above, current research benchmarks are inadequate in the real-world setting of threat intelligence analysis.

**Our work.** This work is conducted within the CTI division of one of the world's leading technology companies, whose platforms and services reach billions of users globally and shape the infrastructure of modern computing. Security is heavily invested because it protects the company's business bottom-line. The data and insights we present in this paper are within this real-world context.

Leveraging workflow data from professional security analysts, we construct CyberThreat-Eval, an expert-annotated benchmark. It addresses three key gaps in existing evaluations: (1) We design tasks that reflect real analyst workflows rather than simplified quizzes or Q&A formats; (2) We introduce analyst-centric metrics such as factual accuracy, time spent, and token cost; (3) We evaluate performance across all three stages of the CTI workflow–triage, deep search, and TI drafting–in an end-to-end manner. We assess four LLMs: two base models (GPT-4o, o3-mini) and two models fine-tuned on a CTI-specific corpus. Results show that while LLMs demonstrate strong recall in triage, they suffer from low precision. In deep search, base models (GPT-4o, o3-mini) retrieve more useful URLs. During TI drafting, IoC extraction reveals a trade-off between speed and recall, and all models struggle with accurately extracting and mapping MITRE ATT&CK TTPs. However, LLMs are notably better at generating coherent root-cause narratives. These findings suggest that current LLMs are effective at retrieving relevant information but need significant improvements in triage precision, TTP reasoning, and cost efficiency. To address these challenges, the CTI workflow needs to utilize external knowledge databases and human experts in the loop. We will explain how these elements are built into our workflow, namely TRA (Figure 3). We will release both CyberThreat-Eval and TRA to support the community in advancing analyst-oriented CTI automation.

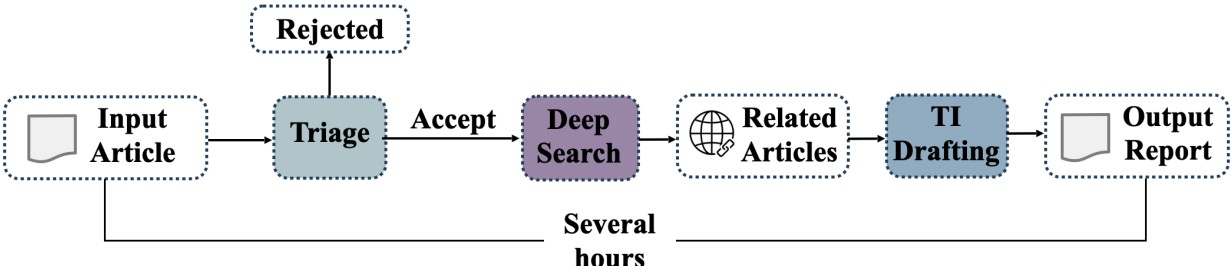

Figure 1: Illustration of the threat analysis workflow.

## 2  Background and Related Works

### 2.1  End-to-end Threat Research Workflow

To effectively process OSINT for threat analysis, a structured end-to-end workflow is essential to handle the scale of cyber threats. As illustrated in Figure 1, a typical production-level workflow transforms raw data feeds into actionable, customer-ready threat intelligence through three distinct stages: triage, deep search, and TI drafting. The process begins with data ingestion, where a crawler collects articles. These articles then enter the triage stage, where each is assessed based on its subject matter and target to determine its relevance and potential impact. Articles that do not meet the criteria are rejected and removed from the workflow. Accepted articles are assigned an incident priority to guide the allocation of analytical resources. For each prioritized incident, the workflow continues into two parallel yet interconnected stages: deep search and TI drafting. In the deep search stage, analysts generate related queries for the incident and follow hyperlinks from the seed article. They gather materials from open-web indices, commercial feeds, and public or proprietary knowledge bases, evaluating each document for relevance and quality. This results in a curated list of URLs that enrich the original case file. In the TI drafting stage, analysts create a working report based on the seed article, iteratively incorporating evidence obtained from the deep search. Elements such as IoCs, MITRE ATT&CK TTPs, timelines, and mitigations are added. The narrative is refined each time new, high-value information is discovered. Iteration ends when further searches no longer yield substantive additions, resulting in a structured, publication-ready report. The final output is a comprehensive, customer-ready article containing key intelligence components. Despite ongoing research and industry efforts to improve these workflows, challenges such as incomplete coverage and inconsistent data quality persist Jin et al. (2024).

### 2.2  LLM Benchmarks for OSINT

Recent research has leveraged LLMs to automate a variety of CTI tasks. As summarized in Table 1, these methods predominantly focus on specific CTI subtasks. For example, IntelEX Xu et al. (2024) extracts TTPs and threat actors from unstructured threat reports, while Cylens Liu et al. (2025) specializes in threat actor profiling and malware clustering. Other approaches, such as HPTSA Fang et al. (2024), identify zero-day vulnerabilities from forum discussions, and CASEY Torkamani et al. (2025) automatically identifies software weaknesses (CWEs) and assesses their severity from codebases. Existing benchmarks such as CTIBench Alam et al. (2024), CTISum Peng et al. (2024), SECURE Bhusal et al. (2024), and CyberMetric Tihanyi et al. (2024) assess performance in threat research tasks and cover a broad range of challenges from knowledge-based question answering to threat actor attribution and vulnerability classification. SEvenLLM-Bench Ji et al. (2024) defined a targeted evaluation with 1,300 test questions covering both English and Chinese threat intelligence content. Although each method demonstrates strengths in isolated tasks, they typically fail to offer comprehensive solutions spanning multiple CTI tasks simultaneously.

When considering coverage across a realistic threat intelligence workflow, consisting of triage, deep search, and TI drafting stages, most existing LLM-based approaches exhibit significant limitations (see Table 2).The reviewed methods reveals systemic gaps. Specifically, of the surveyed methods, only HPTSA Fang et al.

(2024) explicitly includes capabilities for both triage and deep search, and only CyberForumCTI Clairoux-Trepanier et al. (2024) covers both triage and TI drafting stages comprehensively. Notably, no single approach adequately addresses all three stages simultaneously. This fragmented coverage prevents these methods from fully automating the end-to-end threat intelligence workflow critical for operational effectiveness.

Beyond task and workflow coverage, current benchmarks and approaches employ model-centric metrics (e.g. ROUGE Lin (2004), BERTScore Zhang et al. (2019), Exact-Match) that reward lexical overlap yet overlook analyst-centric costs such as token usage and latency. Furthermore, task formats are often quiz-style (like multi-choice questions, Q&A), which do not mirror how practitioners triage incidents or compile intelligence briefs.

Table 1: Existing CTI benchmarks and approaches: tasks and metrics

| Benchmark | Task Scenario | Reported Metrics |
|---|---|---|
| CTIBench Alam et al. (2024) | CTI Context Understanding and Analyzing | Accuracy; Macro-$F_1$; Exact-Match; MAD |
| CTISum Peng et al. (2024) | Attack Process Summarization | ROUGE-1/2/L; BERTScore; Human A/B Ratings |
| SEvenLLM-Bench Ji et al. (2024) | Incident Analysis (like Vulnerability Exploitation Analysis, Attack Intent Analysis, ...) and Understanding (like Cybersecurity Event Classification, Malware Extraction, ...) | Exact-Match; Micro/Macro-$F_1$; ROUGE-L; BLEU |
| CyberMetric Tihanyi et al. (2024) | Cybersecurity Knowledge Q&A | Accuracy; Human vs. Model Accuracy |
| SECURE Bhusal et al. (2024) | Cybersecurity Knowledge Extraction and Reasoning | Accuracy; ROUGE-L; MAD |
| CyberBench Liu et al. (2024b) | Threat Report Summarization; Threat-related QA and Category Classification | ROUGE; Accuracy; Micro/Macro-$F_1$ |
| HPTSA Fang et al. (2024) | Identification of Recent LLM-Unseen Vulnerabilities | Overall Success Rate |

Table 2: Workflow-stage coverage of existing benchmarks and approaches

| Benchmark | Triage | Deep Search | TI Drafting | | | |
|---|---|---|---|---|---|---|
| | | | IoCs | TTPs | Threat Actors | Root Cause |
| CTIBench Alam et al. (2024) | ✗ | ✗ | ✗ | ✓ | ✓ | ✓ |
| CTISum Peng et al. (2024) | ✗ | ✗ | ✗ | ✗ | ✗ | ✗ |
| SEvenLLM-Bench Ji et al. (2024) | ✗ | ✗ | ✓ | ✓ | ✓ | ✓ |
| CyberMetric Tihanyi et al. (2024) | ✗ | ✗ | ✗ | ✗ | ✗ | ✗ |
| SECURE Bhusal et al. (2024) | ✗ | ✗ | ✗ | ✓ | ✗ | ✓ |
| CyberBench Liu et al. (2024b) | ✓ | ✗ | ✗ | ✓ | ✓ | ✓ |
| HPTSA Fang et al. (2024) | ✗ | ✗ | ✗ | ✗ | ✗ | ✗ |

## 3 CyberThreat-Eval

### 3.1 Overview

As outlined in the Introduction (Sec 1), existing benchmarks for CTI exhibit several critical limitations. To address these gaps and provide a more practical and comprehensive framework for evaluating LLMs in automated CTI research, we present the CyberThreat-Eval benchmark. Our contributions are threefold. First, we design practical tasks that closely mirror real-world analyst workflows. Second, we implement analyst-centric evaluation methods that go beyond lexical metrics to emphasize practical utility and real-world relevance. Third, we enhance workflow coverage by targeting three critical stagestriage, deep search, and threat intelligence (TI) draftingwith task designs tailored to each stage. Our CyberThreat-Eval aims to

establish a rigorous, end-to-end standard for assessing an LLMs ability to perform the complex, high-value tasks essential to security analysts.

## 3.2 Data Construction

To ensure that our benchmark reflects the practical challenges faced by security professionals, we construct it using representative data sources drawn from real-world, industry-grade threat intelligence operations. Each task is designed to evaluate the extraction and classification of threat-related information, with ground-truth annotations provided by security research experts. Table 3 summarizes the data distribution across the three primary workflow stages. For the triage stage, experts assigned priority scores to 488 articles. The deep search component is evaluated using an initial set of 55 input URLs. The TI drafting stage, which involves more complex and multi-faceted tasks, is assessed through several representative evaluations of core capabilities. In the IoC extraction task, ground-truth answers are straightforward to identify, as they consist of concrete, structured artifactssuch as file hashes, IP addresses, or domain namesthat clearly indicate malicious activity. In contrast, the TTP assignment task requires expert judgment: human annotators manually review articles to identify and label relevant Tactics, Techniques, and Procedures (TTPs), evaluate their alignment with the incident's subject and target, and assign an appropriate priority score. For the content generation evaluation, 412 articles were selected by experts for their richness in analytical depth. This evaluation specifically tests the LLMs' ability to generate coherent and informative narratives about the threat actor and root cause of each incident. These two elements were chosen because they represent some of the most cognitively demanding aspects of threat analysisrequiring models to go beyond extraction and synthesize the who and why behind an incident. To support focused evaluation, some tasks distinguish between two types of inputs: "description" (a concise summary of the incident) and "article" (the full original source).

## 3.3 Task Description

This benchmark evaluates LLMs through a series of tasks that reflect the end-to-end workflow of cyber threat research. The tasks are organized into three key stages: triage, deep search, and TI drafting. Each task is designed to assess the specific skills required for effective and comprehensive threat analysis at each stage

Table 3: Data Description for Benchmark Tasks by Workflow Stage.

| Category | Task / Component | Data Size |
|---|---|---|
| Triage | Triaging & Assigning Priority Scores | 488 articles |
| Deep Search | Identifying Related Articles | 55 articles |
| TI Drafting | IoCs Extraction | 1310 IoCs |
| | TTPs Identification | 1565 TTPs |
| | Threat Actor Generation | 412 articles |
| | Root Cause Generation | 412 articles |

### 3.3.1 Triage

This stage evaluates the LLMs' ability to prioritize incidents using only readily available information without extensive analysis. To systematically measure their effectiveness, we design a two-part evaluation focusing on incident prioritization based on subject matter (e.g., mobile malware) and the affected target (e.g., a single system). Effective prioritization requires synthesizing multiple dimensions of threat intelligence, guiding analysts to identify and address the most critical incidents first. Specifically, we define two tasks for evaluation:

**Task 1: article triage**. In this task, LLMs are asked to determine whether an article should be *"accepted"* (i.e., if it provides valid, actionable subject and target information about a threat incidentsufficient to support identifying threat actors, root causes, etc.) or *"rejected"* (if it lacks relevant or actionable insights) for further analysis.
**Task 2: priority score assignment**. For articles in the *"accepted"* category, the task is to assign an appropriate priority score that reflects the severity and urgency of the incident. LLMs are evaluated based on their ability to classify articles in Task 1 and to assign accurate priority scores in Task 2 according to predefined criteria.

Human expert judgments are used to validate the scoring system and assess the reliability of the LLMs' prioritization. These scores help analysts assess the relative importance of each incident and prioritize their responses accordingly. The detailed scoring criteria are provided in Appendix B.

### 3.3.2 Deep Search

Following triage, analysts typically collect additional information to deepen their understanding of an incident. The deep search evaluation assesses the capability of LLMs to gather and analyze relevant external resources that provide valuable context about a particular threat. Specifically, given an initial input URL describing a cybersecurity incident, the LLM is tasked with searching the web to identify additional URLs that contain supplementary or supporting details about the same incident. To evaluate the effectiveness of this retrieval process, we employ an LLM-based comparative analysis. For each candidate URL, its content is systematically compared against the original seed article. An LLM, acting as a cybersecurity expert, is prompted to determine if the retrieved article contains genuinely new and valuable "additional information," such as novel facts, data points, different analytical perspectives, or extra technical details that are absent from the original source. The evaluating LLM outputs a structured decision ('true' or 'false'), allowing us to quantify the "average URLs with additional information" and thereby measure the quality and utility of the deep search results.

### 3.3.3 TI Drafting

The final stage assesses the model's ability to synthesize information (from the initial article and potentially from the deep search stage) into structured and insightful threat intelligence reports. To measure the effectiveness, we design the following tasks below, which are categorized into two main areas.

**(1) Understanding and reasoning of key threat elements.** This section evaluates how well LLMs can extract and interpret core elements of cyber threat intelligence from unstructured articles. It focuses on two key types of information: IoCs and MITRE ATT&CK TTPs.

**IoC extraction.** This task assesses the LLMs' ability to accurately identify and extract explicit IoCs from cybersecurity articles. IoCs are essential for identifying malicious activity, attack vectors, and compromised systems. The core objective is to evaluate the LLMs' proficiency in recognizing and extracting these technical indicators from text.

**MITRE TTP identification and mapping.** This task evaluates LLMs' ability to automatically detect adversary behaviors described in incident reports and map them to the appropriate TTPs within the MITRE ATT&CK framework. Accurate TTP identification is critical for understanding how threat actors operate and for designing effective defense strategies. LLMs must process each article to identify and associate relevant TTPs, which is more challenging than IoC extraction.

**(2) Contextual intelligence evaluation based on human expert criteria.** This area evaluates the ability of LLMs to generate detailed, accurate, and coherent narrative content, providing analysts with deeper insights into incidents. It comprises two primary tasks: (1) producing descriptive profiles of identified threat actors by synthesizing article context with broader domain knowledge, and (2) creating detailed narratives on the incident's root cause, including attack mechanisms, exploited vulnerabilities, and event timelines. Outputs are evaluated by human experts on relevance, accuracy, comprehensiveness, clarity, coherence, and proper source attribution. The goal is to produce intelligence summaries significantly richer and more actionable than the original sources alone.

## 4 Experimental Results

### 4.1 Experiment Setup

We evaluate diverse LLMs representing varying capabilities and deployment scenarios. For general-purpose models, we select the pre-trained GPT-4o and the reasoning-oriented o3-mini. Additionally, we fine-tune GPT-4o and GPT-4o-mini on analyst-triaged data from 2024, denoted respectively as `GPT-4o (FT)`

(`fine-tuned gpt-4o-2024-08-06`) and GPT-4o-mini (FT) (`fine-tuned gpt-4o-mini-2024-07-18`). We employ a standard Supervised Fine-Tuning (SFT) approach designed to bolster the LLMs' overall security-related capabilities across diverse CTI tasks. The SFT dataset is a curated aggregation of multiple high-quality, proprietary datasets derived from our security operations. This multi-task corpus encompasses a variety of CTI-related functions, including but not limited to Question Answering (QA), CTI Mapping, Natural-Language-to-Query Translation, and Function Calling. For rigorous monitoring of the fine-tuning process, the aggregated dataset is partitioned into training (79%), validation (1%), and testing (20%) sets. All models operate at the temperature of 0.01 and seed as 42 for consistency. To evaluate narrative quality (threat actors and root causes), we employ the LLM-as-Judge paradigm Liu et al. (2023); Verga et al. (2024). This evaluation method provides the LLM judge with the ground-truth article, candidate output, and a scoring rubric covering relevance, factual accuracy, comprehensiveness, clarity, coherence, and attribution. Each dimension is rated on a 1-5 scale. To ensure the reliability of our LLM-as-Judge evaluation framework, we collaborate with senior security experts to draft rule-based criteria, requiring the judge LLM to output a justification field alongside its decision. Subsequently, we conducted a validation study using a diverse set of approximately 100 article examples. Human experts review the LLM's judgments and provide detailed qualitative feedback on instances where the LLM's reasoning deviate from expert consensus. Based on this feedback, we systematically refine the evaluation details and prompt instructions over two rounds of testing. This calibration process yields an agreement rate between the LLM-as-Judge and human experts exceeding 95%. This high level of agreement shows that the LLM-as-Judge can be a reliable method for expert evaluation in this specific domain.

Metrics align explicitly with workflow stages. For triage, models perform two sequential tasks: (i) deciding whether an article should be accepted (measured by precision and recall), and (ii) assigning a priority score to accepted articles. Priority assignment performance is quantified by pass rate (percentage of exact matches with analyst-provided scores) and average bias (mean signed deviation from ground truth). Operational efficiency is assessed via processing time and token consumption per article. In the deep search stage, we measure: the average number of newly discovered URLs per incident, and the subset of URLs providing additional information (e.g., IoCs, exploit code, or mitigations). Efficiency metrics (processing time and tokens used) are also recorded. In the TI drafting stage, two task categories are evaluated. For IoC extraction and TTP identification, we report precision, recall, latency, and token usage. For generating threat actor and root cause narratives, we apply the six-dimension evaluation rubric mentioned earlier to comprehensively assess content quality.

## 4.2 Overall Results

This section presents the detailed experimental results in our comprehensive threat research benchmark for each stage of the workflow. Each subsection elaborates on a specific task, and describing the performance of various LLMs. The result table is shown in Table 4.

**Triage performance.** Triage evaluation focuses on two components: article classification ("accepted" or "rejected") and priority score assignment. Results are shown in Table 4. For article classification, LLMs exhibit high recall (often >0.90), indicating strong sensitivity to potentially relevant articles. However, precision remains low (typically <0.40), reflecting a tendency to over-accept, which may increase analysts workload. For priority scoring, models achieve 55-60% accuracy, with fine-tuned variants like GPT-4o (FT) and GPT-4o-mini (FT) occasionally aligning more closely with expert judgments. Efficiency varies significantly: GPT-4o is the fastest and most token-efficient, while o3-mini and fine-tuned models consume more time and tokens, especially when processing full-article inputs. These results highlight a trade-off between accuracy and operational costLLMs can broadly identify relevant content but need improvement in precision and more nuanced prioritization.

**Deep search performance.** The deep search stage evaluates LLMs' ability to retrieve and rank contextual information that enriches incident understanding. As shown in Table 4, base models like GPT-4o and o3-mini process more URLs on average compared to fine-tuned models. They also identify more URLs with additional relevant information (3.54 and 3.00, respectively). In contrast, fine-tuned LLMs are much more conservative. This suggests that fine-tuning may lead to more focused retrieval, possibly because these

Table 4: Performance Metrics for Vanilla LLMs by Task Category, Task, Metric, and Model

| Category | Task | Metric | GPT-4o | o3-mini | GPT-4o (FT) | GPT-4o-mini (FT) |
|---|---|---|---|---|---|---|
| Triage | Priority Scoring (Description) | Precision (Accepted) | 0.3590 | 0.3982 | 0.3392 | 0.3477 |
| | | Recall (Accepted) | 0.9899 | 0.9091 | 0.9798 | 0.9798 |
| | | Pass Rate (Score) | 0.566 | 0.550 | 0.600 | 0.590 |
| | | Avg. Bias (Score) | 0.505 | 0.680 | 0.470 | 0.520 |
| | | Total Time (s) | 708.23 | 5178.38 | 3939.35 | 4424.71 |
| | | Total Tokens | 430124 | 450093 | 453280 | 555547 |
| | Priority Scoring (Article) | Precision (Accepted) | 0.3037 | 0.3802 | 0.2717 | 0.2964 |
| | | Recall (Accepted) | 1.0000 | 0.9293 | 0.9798 | 1.0000 |
| | | Pass Rate (Score) | 0.566 | 0.580 | 0.590 | 0.600 |
| | | Avg. Bias (Score) | 0.505 | 0.550 | 0.500 | 0.450 |
| | | Total Time (s) | 785.64 | 6048.47 | 6034.71 | 7283.13 |
| | | Total Tokens | 1438448 | 1390052 | 1522969 | 1891248 |
| Deep Search | URLs Extraction | Avg. URLs processed (w. input) | 6.22 | 4.93 | 1.75 | 1.25 |
| | | Avg. URLs w. additional info | 3.54 | 3.00 | 0.38 | 0.22 |
| | | Avg. Processing Time(s) | 484.94 | 322.95 | 950.68 | 345.24 |
| TI drafting | IoC Extraction | Precision | 0.8240 | 0.8503 | 0.8846 | 0.6944 |
| | | Time (s) | 580 | 7535 | 727 | 808 |
| | | Tokens | 339080 | 448152 | 331766 | 319700 |
| | MITRE ATT&CK TTP Identification | Precision | 0.2787 | 0.3480 | 0.2387 | 0.1771 |
| | | Recall | 0.2270 | 0.1759 | 0.1846 | 0.1414 |
| | | Time (s) | 818 | 5097 | 2254 | 1889 |
| | | Tokens | 466426 | 432593 | 453282 | 493860 |
| | Content: Threat Actor | Relevance | 1.547 | 3.964 | 3.964 | 2.475 |
| | | Accuracy | 1.528 | 3.656 | 3.655 | 2.405 |
| | | Comp. | 1.145 | 3.165 | 3.165 | 1.755 |
| | | Clarity | 2.019 | 4.753 | 4.752 | 3.699 |
| | | Coherence | 1.734 | 4.731 | 4.731 | 3.405 |
| | | Attribution | 1.140 | 2.968 | 2.967 | 1.832 |
| | Content: Root Cause | Relevance | 3.686 | 4.733 | 4.533 | 3.808 |
| | | Accuracy | 3.458 | 4.627 | 4.371 | 3.819 |
| | | Comp. | 3.362 | 4.576 | 4.286 | 3.441 |
| | | Clarity | 3.932 | 4.818 | 4.629 | 4.163 |
| | | Coherence | 3.753 | 4.800 | 4.586 | 3.988 |
| | | Attribution | 3.612 | 4.418 | 4.323 | 3.667 |

GPT-4o (FT) refers to Finetuned GPT-4o-2024-08-06; GPT-4o-mini (FT) refers to Finetuned GPT-4o-mini-2024-07-18.
Comp.: Comprehensiveness. Time (s) and Tokens are rounded. Content scores (Relevance, Accuracy, Comp.) are averages.

models possess greater inherent knowledge of specific incidents due to specialized training. As a result, they may perform less external searching or consider fewer external sources as offering truly novel insights.

**TI drafting performance**    This stage evaluates the LLMs' ability to synthesize information from given articles into structured, insightful threat intelligence components. Our findings, summarized in Table 4, indicate notable variability in LLM performance across drafting tasks, broadly categorized into key element extraction and narrative content generation.

*Key elements extraction.* Extracting fundamental threat elements like IoCs and MITRE ATT&CK TTPs reveals distinct LLM strengths and weaknesses. Base models like GPT-4o and o3-mini demonstrate strong precision (around 0.82-0.85). However, efficiency is a concern: for example, while o3-mini is accurate, it is significantly more resource-intensive in terms of both time and token usage compared to GPT-4o. Smaller models, such as GPT-4o-mini (FT), perform worse in precision (0.6944), highlighting that model size and fine-tuning are critical for reliable and cost-effective IoC extraction. In contrast, the TTP identification task remains a substantial challenge. All evaluated vanilla LLMs exhibit low precision and recall (mostly below 0.35). Although o3-mini achieves the highest TTP precision (0.3480), it also has the lowest recall, indicating it misses many relevant TTPs. Fine-tuning does not consistently improve performance, suggesting that this task demands deeper reasoning about adversarial behavior than current LLMs are typically capable of.

*Content generation quality.* When generating descriptive narratives, LLM performance varies significantly between explaining threat actors and identifying root causes. For threat actor content, base GPT-4o performs poorly. In contrast, o3-mini and GPT-4o (FT) produce markedly better results, with high relevance and

accuracy scores (around 3.7-4.0). However, their ability to provide fully comprehensive and well-attributed details (comprehensiveness and attribution scores around 3.0) still needs improvement. For root cause content, LLMs generally perform better, achieving acceptable scores across all six evaluation criteria. This suggests that LLMs are currently more adept at explaining the more factual "how" of an incident (root cause) than the often more inferential and complex "who" (threat actor). Their ability to generate clear and logical root cause analyses stands out as a notable strength.

### 4.3 Results Summary and Analysis

The experimental results highlight both the promising capabilities and current limitations of LLMs in automated cyber threat research tasks.

**Performance varies in different tasks.** In the triage stage, LLMs exhibit strong recall capabilities, consistently retrieving most relevant articles; however, they frequently struggle with precision, accepting many irrelevant entries. During the deep search stage, general-purpose LLMs such as GPT-4o and o3-mini typically identify more URLs containing valuable additional information compared to their fine-tuned counterparts. This implies that fine-tuned LLMs might be more selective or conservative, potentially due to their specialized training reducing the need for extensive external information searches. Additionally, processing efficiency (measured by retrieval time) varies significantly, with smaller or fine-tuned LLMs occasionally outperforming larger base LLMs. In the TI drafting stage, the extraction of IoCs is relatively robust, although improvements in recall and processing efficiency remain necessary. By contrast, identifying and mapping TTPs presents substantial challenges for all evaluated LLMs. For narrative content generation, LLMs demonstrate notable proficiency in creating clear, coherent explanations of root causes. However, consistently generating comprehensive, well-attributed descriptions of threat actors proves considerably more challenging.

**The performance-cost trade-off challenge.** LLMs consistently exhibit a fundamental trade-off across threat research tasks: enhancing recall (identifying all potentially relevant information, as seen prominently in triage) often diminishes precision (the accuracy of the identified information), simultaneously increasing operational costs (time and token consumption). Conversely, attempts to boost precision typically result in lower recall or demand greater resources. The TTP identification task exemplifies this clearly: the o3-mini achieves the highest precision observed (0.35 compared to GPT-4o's 0.28) but consumes approximately six times more total processing time and 1.3 times more tokens. On the other hand, fine-tuned GPT-4o substantially reduces latency by nearly two-thirds, but its precision correspondingly drops to 0.24. Analysts are thus confronted with a clear operational dilemma: they must either accept slower, resource-intensive processes for enhanced accuracy, or prioritize faster, cost-effective solutions at the expense of potentially overlooking relevant or introducing inaccurate information.

**No one-fits-all solution of fine-tuning (specialization vs. generalization).** Our results indicate that fine-tuning LLMs on domain-specific data can significantly enhance their performance on targeted tasks, such as generating detailed threat-actor profiles. However, this approach does not universally improve all performance metrics and may occasionally degrade certain capabilities. For example, fine-tuning appears to limit LLMs exploratory breadth, making it less effective in tasks requiring extensive external information retrieval, as evidenced in our deep search evaluations. Additionally, fine-tuning alone does not substantially mitigate fundamental reasoning limitations, such as the persistent challenges observed in TTP mapping tasks.

**Poor reasoning on cybersecurity-related knowledge.** LLMs consistently face difficulties when tasks demand deeper reasoning and inference over cybersecurity knowledge. Two specific examples highlight this challenge. First, the identification and mapping of MITRE ATT&CK TTPs require LLMs to infer adversarial intent from subtle contextual clues. Current LLMs frequently miss these nuances or produce inaccurate mappings; for instance, GPT-4o achieves only 0.28 precision and 0.23 recall on our benchmark's TTP identification task. Second, generating comprehensive threat-actor profiles demands synthesizing disparate pieces of incident-related evidence into a coherent narrative. Baseline GPT-4o receives notably low scores

from expert judges in this area, averaging just 1.55 for relevance and 1.14 for comprehensiveness on a five-point scale. The resulting summaries are often superficial and, in worse cases, contain plausible yet incorrect statements (hallucinations). These shortcomings illustrate that merely optimizing for recall or precision is insufficient. Future research must focus on enhancing the model's capability to reason across multiple documents and to rigorously ground every assertion in verifiable evidence, both essential qualities for trustworthy CTI analyses.

### 4.4 Qualitative Analysis: Illustrating LLM Performance Gaps

In addition to quantitative evaluations, qualitative assessments of LLM-generated outputs reveal significant operational shortcomings in realistic threat intelligence scenarios. These limitations underscore the necessity of developing more sophisticated and structured approaches to achieve comprehensive automation of CTI tasks. Specifically, as illustrated in Figure 2, we observe two main types of performance gaps: (1) the generation of sparse or incomplete content for critical intelligence elements, such as threat actor profiles and root cause explanations, when compared to expert expectations; and (2) inaccuracies or hallucinations in technical details, such as misidentifying IoCs or incorrectly classifying MITRE ATT&CK TTPs. These qualitative examples indicate that although LLMs demonstrate foundational competencies, their standalone deployment in complex CTI operations may yield intelligence that is incomplete, imprecise, or potentially misleading. Therefore, there is a clear need for integrated frameworks capable of supplementing LLMs with effective knowledge retrieval, domain-specific reasoning, and rigorous verification methods.

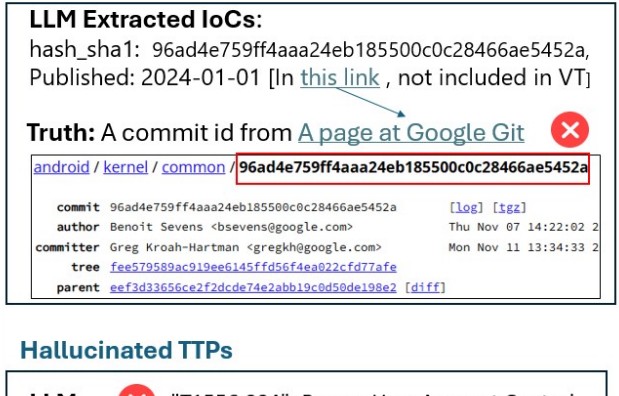

Figure 2: Illustrative examples of LLM performance gaps: sparse content generation for threat actors/root causes, and hallucination of IoCs and TTPs compared to ground truth or expert expectations.

**Absence of domain expert experience.** Figure 2 (left) highlights a recurring shortcoming of vanilla LLMs when they are asked to provide descriptive context for entities such as threat actors or to explain the root cause of an incident. The LLMs frequently produce brief, high-level statements—for example, "WIRTE, an APT group linked to Hamas..." or "QBot malware relies on SVG files..." Although these remarks may be factually correct, they lack depth. Key dimensions that domain experts expect, historical campaigns, typical victim profiles, and broader strategic impact, are absent. Expert reviewers consistently flag this deficit in their feedback. Typical comments include "This is generally accurate but sparse; more context

would be helpful," or "For the root cause, add a subsection that provides additional detail on the malware or tool," when they compare the model output with the richer reference text on the right side of Figure 2. These observations indicate that, without explicit domain conditioning, current LLMs struggle to synthesize scattered technical evidence into a coherent, multi-layered narrative. Consequently, analysts must still invest substantial effort to expand and validate LLM-generated content before it is suitable for professional CTI reporting.

**Absence of external ground truth to verify technical details.** Internal knowledge encoded within LLMs alone is insufficient to reliably verify technical details, such as IoCs or MITRE ATT&CK TTP mappings, due to the complexity, specificity, and rapidly evolving nature of cybersecurity threats. Accurate verification typically requires authoritative external databases (e.g., VirusTotal) that maintain up-to-date and validated intelligence. Without integrating such external sources, LLMs are prone to hallucinate or misinterpret technical artifacts. The top-right example in Figure 2 illustrates an LLM extracting a commit ID ("96ad4e...") and misrepresenting it as a SHA1 hash IoC. This type of errors can mislead analysts onto an incorrect investigative path. Similarly, the bottom-right example shows an LLM incorrectly mapping an observed behavior to a MITRE ATT&CK TTP ("T1556.004: Bypass User Account Control") when the correct mapping is different ("T1556.004: Modify Authentication Process: Network Device Authentication" MITRE Corporation (b)). Such inaccuracies in identifying specific IoCs or TTPs undermine the reliability of LLM-generated intelligence.

## 5  Threat Research Agent (TRA)

### 5.1  Method

Within the CTI division of the company, we have developed the Threat Research Agent (TRA), an end-to-end threat research framework that has been fully integrated into the organization's CTI workflow. TRA operates as an iterative, human-in-the-loop agent system designed to improve information integration, enable streamlined verification, and produce structured, actionable intelligence for analysts. Its architecture, illustrated in Figure 3, emphasizes a continuous cycle of analysis driven by LLMs and subsequently refined through expert feedback. Specifically, TRA starts with an LLM performing preliminary research on a seed article, generating relevant queries, and retrieving supplementary content from various knowledge sources such as public platforms like Bing and Google, as well as proprietary databases. After gathering the relevant information, an LLM-based selector filters these links to ensure they are pertinent to the seed article. This curated content is then assessed by an LLM evaluator to refine the selection further. The LLM synthesizes the refined information into an initial draft, which is specifically structured for human review. By embedding domain expert feedback into the workflow (via prompt engineering), TRA directly addresses the challenge of lack of domain-expert human experience, ensuring that the generated content is enriched with the depth and context required by human analysts. In addition, TRA integrates external authoritative knowledge sources, such as VirusTotal, into the workflow for cross-checking and validating LLM outputs. This integration minimizes hallucinations and ensures that the technical details, such as IoCs and MITRE ATT&CK TTPs, are accurate and reliable, thus enhancing the overall credibility of the threat intelligence.

**Embedding domain-expert feedback.** The TRA framework is designed to seamlessly integrate domain-expert feedback throughout the entire automated threat research process, as depicted in Figure 3. The output draft of TRA, rich with citations and supporting evidence, serves as a dynamic interface where domain experts can easily validate the findings, provide corrections, and suggest further areas of exploration. This human-in-the-loop approach is crucial for ensuring the accuracy, relevance, and completeness of the final intelligence report. As an integral part of this workflow, expert feedback is continuously captured and incorporated into the system, enabling the LLM to refine future outputs. Additionally, multiple verifiers and checkers cross-check the technical details within the report, such as IoCs and TTPs, ensuring they align with authoritative external data sources. This expert-driven iterative cycle improves the overall quality of the intelligence generated, addressing the gaps in LLMs' ability to fully synthesize complex cybersecurity information.

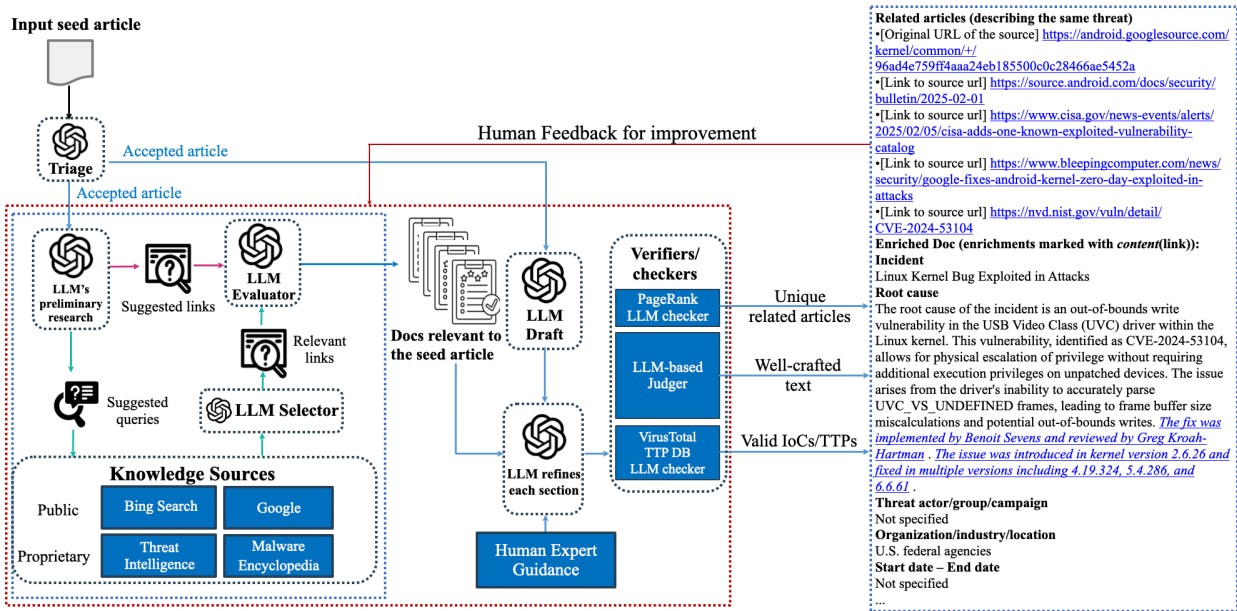

Figure 3: The illustration of how TRA works

**Integrating external ground truth.** As mentioned in Section 4.4, a key limitation of LLMs in cybersecurity analysis is their inability to reliably verify technical details such as IoCs and MITRE ATT&CK TTPs. This limitation arises from the internal knowledge constraints of LLMs, which often prevent them from accurately handling such tasks on their own. To address this, we integrate authoritative external knowledge sources like VirusTotal into our workflow. The verifier/checker component cross-checks and validates LLM outputs, correcting misidentified IoCs or TTPs, refining generated content, and adjusting priority scores. This process ensures that the information provided by the LLMs aligns with trusted, up-to-date data sources, thereby minimizing the occurrence of hallucinations and enhancing the accuracy of the intelligence.

## 5.2 TRA Results Analysis

TRA is used by the analysts in our team, and the outputs it generates are reviewed by a team of experts. Their detailed feedback, including corrections and refinements, is systematically collected. The additional feedback captured by TRA further automates threat research. A representative example illustrates this improvement clearly. TRA successfully highlighted that many MikroTik devices shipped with a hardcoded admin account and a blank password that were exploited by threat actors, a critical fact initially overlooked by the human analyst. The expert reviewing this case remarked, *"I was impressed that the model surfaced the fact that 'Many MikroTik devices shipped with a hardcoded admin account with a blank password, which was exploited by the threat actors.' I actually read several articles myself but did not surface this detail."* Additionally, experts noted that integrating IoCs, detections, mitigations, and MITRE ATT&CK TTPs into a single, cohesive report significantly reduced the need for analysts to switch between multiple tools, thus making the final intelligence "more coherent than the original articles." This feedback underscores TRA's practical value in day-to-day workflows. Detailed examples and expert comments illustrating this iterative feedback loop are provided in Appendix E.2. The expert review process not only validates the quality of individual outputs but also generates a valuable dataset of human-labeled examples, facilitating the alignment of the underlying models more closely with expert judgment.

TRA's performance is further rigorously evaluated using the CyberThreat-Eval benchmark. The quantitative results presented in Table 6 (Appendix D) clearly illustrate the benefits of our human-in-the-loop approach. Specifically, TRA improves IoC extraction precision by 26 percentage points across all base LLMs. For the more challenging TTP identification task, TRA raises precision significantly: from 0.28 to 0.42 for o3-mini and from 0.24 to 0.31 for fine-tuned GPT-4o. Additionally, TRA reduces latency for GPT-4o

by approximately 70 seconds, demonstrating gains in both quality and efficiency. The most substantial improvements appear in content generation for threat actor narratives, where expert evaluations of relevance, clarity, and coherence consistently rise above 4.5 on a five-point scale. In practical terms, TRA effectively transforms initial, sparse drafts into content that experts regard as "publish-ready." Nevertheless, TRA currently retains limitations, as it cannot yet fully automate the entire threat-research workflow without human oversight.

# 6 Conclusion

This paper introduces CyberThreat-Eval, a novel benchmark designed for a holistic, practical, and analyst-centric evaluation of LLMs in automated threat intelligence research. CyberThreat-Eval incorporates tasks closely aligned with real-world analyst workflows and employs human-centric assessments alongside cost-aware performance metrics. Our evaluation reveals that while LLMs can substantially reduce the workload of human analysts, achieving expert-level performance in specific areas remains challenging. In particular, the delicate balance between recall and precision during the triage stage continues to pose difficulties, and tasks requiring complex reasoning, such as identifying and mapping MITRE ATT&CK TTPs, remain significant hurdles. Furthermore, we introduce the Threat Research Agent (TRA), an end-to-end framework that produces human-verifiable outputs. Case studies highlight TRA's effectiveness in supporting analysts through automated threat research, and it has been successfully integrated into the company's daily CTI workflow. Future work will focus on expanding the CyberThreat-Eval benchmark and further enhancing the robustness and efficiency of the TRA framework.

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

# A    Background

## A.1   OSINT

Open Source Intelligence (OSINT) refers to the collection and analysis of information from publicly available sources for intelligence purposes. In cyber threat analysis, OSINT encompasses data from open web content, social media, technical blogs, hacker forums, dark web marketplaces, and other public repositories that can reveal IoCs or adversary MITRE ATT&CK TTPs MITRE Corporation (a). By leveraging OSINT, organizations can quickly gather IoCs Microsoft Corporation (2025a) (such as IP addresses, domain names, malware signatures, etc.) and strategic insights (e.g., threat actor profiles Microsoft Corporation (2025b) or root causes) that would be difficult to obtain through proprietary sources alone. However, as the volume of data increases, effectively harnessing OSINT for threat intelligence presents a significant challenge. Analysts must address issues related to the validity and credibility of a vast amount of data when using OSINT, along with ethical and legal considerations surrounding the monitoring of public information. Therefore, careful filtering and validation to separate credible signals from noise have become critical tasks.

## A.2   LLM for Threat OSINT

LLMs present new opportunities for handling OSINT data at scale. Trained on vast corpora and capable of understanding and generating human language, LLMs are well-suited for tasks like processing unstructured text, extracting insights, and producing summaries or reports. In the OSINT context, researchers have begun to leverage LLMs for various tasks: for example, classifying whether social media posts indicate cybersecurity threats, extracting entities (e.g., names, IoCs) from technical reports, or translating and summarizing foreign news articles. Recent studies show that several LLM-based chatbots, when applied to Twitter cybersecurity data, can achieve an F1 score of up to 0.94 on a threat-related classification task Shafee et al. (2024). LLMs can reliably distill noisy OSINT sources (such as illicit forums) into structured intelligence, greatly accelerating what analysts could do manually Clairoux-Trepanier et al. (2024). Beyond classification and extraction, LLMs are also being used to synthesize and generate threat intelligence reports Perrina et al. (2023). As research in this area continues to progress, promising improvements in performance are emerging. However, while LLMs offer significant potential for OSINT applications, they are not without limitations. For example, LLMs can produce erroneous or unverified information (hallucinations) and often lack up-to-date knowledge in OSINT Li et al. (2025). Thus, in addition to simply applying an LLM, domain adaptation and human oversight are necessary for effective OSINT analysis and generation. Moreover, there is a need for more granular prompt engineering and validation to ensure that LLM-driven intelligence is both accurate and trustworthy.

# B    Scoring Criteria

LLMs accelerate the evaluation process and reduce the need for extensive manual analysis. To better quantify incident priority and further ease the workload for analysts, we introduce a scoring mechanism that maps threat context to numerical values. The underlying principle is that a lower numerical score corresponds to a higher priority. A score of 1 indicates the highest-priority category, while a score of 5 indicates that the threat category is not considered to exist. Numeric values are assigned based on specific attributes of the threat and its target. The scoring criteria are presented in Table 5. In the table, the horizontal columns represent the target categories of the incidents, and the vertical rows represent the subject matter categories of the incidents.

Table 5: Priority Scoring for Threat Categories

| Threat Category | Unknown/NA | Singular System | Singular Company | Singular Country | Multiple Countries |
|---|---|---|---|---|---|
| Defacement / Spam | 5 | 3 | 3 | 3 | 3 |
| Mobile Malware | 3 | 3 | 3 | 3 | 3 |
| Malware Updates | 2 | 3 | 3 | 3 | 2 |
| New Malware | 3 | 3 | 3 | 3 | 2 |
| Vulnerability Exploitation (CVE < 9) | 5 | 2 | 2 | 5 | 5 |
| Cryptominer / Resource Hijacking | 3 | 3 | 3 | 3 | 3 |
| Phishing Campaign | 2 | 2 | 2 | 1 | 1 |
| 0-Day Vulnerability Exploitation | 5 | 1 | 1 | 5 | 5 |
| Vulnerability Exploitation (CVE $\geq$ 9) | 5 | 1 | 1 | 5 | 5 |

| Threat Category | Industry/Sector | Platform/Service | Drive-by | ICS(Industrial Control System) |
|---|---|---|---|---|
| Defacement / Spam | 3 | 3 | 3 | 1 |
| Mobile Malware | 3 | 3 | 3 | 5 |
| Malware Updates | 2 | 2 | 2 | 1 |
| New Malware | 2 | 2 | 2 | 1 |
| Vulnerability Exploitation (CVE < 9) | 5 | 2 | 5 | 1 |
| Cryptominer / Resource Hijacking | 3 | 2 | 3 | 1 |
| Phishing Campaign | 1 | 1 | 2 | 1 |
| 0-Day Vulnerability Exploitation | 5 | 1 | 5 | 1 |
| Vulnerability Exploitation (CVE $\geq$ 9) | 5 | 1 | 5 | 1 |

# C  Content Evaluation

## C.1  Deep Search Evaluation

The deep search stage evaluates the LLMs' ability to retrieve relevant external information to enrich the understanding of an incident. A key component of this evaluation involves assessing whether the retrieved URLs offer genuinely new and valuable information beyond that contained in the initial seed article. To quantify this, we employ an LLM-based comparative analysis. For each URL retrieved by a model (referred to as the "comparison blog"), its content is systematically compared against the original seed article (the "reference blog"). The evaluation prompt is as follows:

```
You are an expert in cybersecurity and information analysis. You are tasked with comparing the
content of two blogs: "reference blog" and "comparison blog".
Task: decide whether the comparison blog contains "additional information" beyond the reference
blog. "Additional information" = any NEW, concrete content that increases understanding, such as:
    - New facts or data (numbers, dates, CVEs, malware names, actors, tools, geography, ...).
    - New analysis or interpretations (novel links, causes, trends, forecasts, mitigation advice,
...).
    - Extra technical or contextual details (methods, background, case studies, impact elaboration,
...).
    - Any other additional details absent from the "reference blog".
Provide your answer as either "True", (the comparison blog has additional information) or "False"
(it does not). If the answer is "True", briefly justify your decision by listing examples of the
additional information found in the comparison blog.
Your response MUST be in valid JSON format with these fields:
{"has_additional_info": true/false, "justification": "Your justification if has_additional_info is
true"}
```

## C.2  Content Evaluation

**Content evaluation criteria.** In this subsection, we introduce the criteria used to evaluate the generated content for the threat actor and root cause. These criteria are essential to ensure that the generated context is accurate, relevant, and comprehensive. Each criterion helps assess a specific aspect of the content's quality. The full set of criteria is shown below.

**Evaluation details.** We evaluate the results using GPT-4o based on the criteria provided for each task (e.g., threat actor, root cause). The temperature is set to 0.01, and the output format is specified as JSON. The output

template is: "Relevance": <score>, "Accuracy": <score>, "Comprehensiveness": <score>, "Clarity": <score>, "Coherence": <score>, "Attribution": <score>. For each evaluation method, we calculate the average score for each criterion by taking the mean of the individual scores across all entries.

## D TRA Evaluation Results

This section illustrates the detailed TRA results from Section 5.2. Table 6 presents a comprehensive comparison between vanilla LLMs and their counterparts enhanced with TRA in multiple tasks, including IoC extraction, TTP identification, and content generation for the description of threats and root causes. A key insight is that TRA's modular architecture systematically enhances LLM outputs for CTI tasks. For IoC extraction, TRA consistently boosts precision over vanilla LLMs (e.g., finetuned GPT-4o: 0.9034 TRA vs. 0.8846 vanilla), although with a trade-off in recall and increased operational costs. This highlights TRA's verification strength, especially on "hard" articles where it achieved near-perfect precision and high recall. In TTP identification, a more challenging area, TRA generally provides modest precision gains and can improve efficiency (e.g., TRA with GPT-4o was faster and used fewer tokens than vanilla GPT-4o). The most significant impact of TRA is observed in content generation quality. For both threat actor and root cause narratives, TRA substantially elevates all six expert-evaluated criteria (relevance, accuracy, comprehensiveness, clarity, coherence, attribution) compared to vanilla outputs across all tested base LLMs. For instance, TRA with o3-mini for root cause analysis achieved average scores often exceeding 4.5, underscoring its ability to synthesize richer, more accurate, and coherent intelligence. Although small decreases in comprehensiveness are present for some TRA configurations, the overall trend confirms the strong capability of TRA in producing actionable and reliable threat intelligence.

Table 6: Performance Overview: Vanilla LLMs vs. TRA (with Corresponding Base LLMs) on Key Metrics

| Task Category | Metric | GPT-4o | | o3-mini | | GPT-4o (FT) | | GPT-4o-mini (FT) | |
|---|---|---|---|---|---|---|---|---|---|
| | | Vanilla | TRA | Vanilla | TRA | Vanilla | TRA | Vanilla | TRA |
| IoC Ext. | Precision | 0.8240 | 0.8496 | 0.8503 | 0.8864 | 0.8846 | 0.9034 | 0.6944 | 0.7469 |
| | Time (s) | 580.13 | 2285.65 | 7534.92 | 9517.04 | 726.95 | 1938.19 | 808.67 | 1966.16 |
| | Tokens | 339080 | 458093 | 448152 | 599989 | 331766 | 452677 | 319700 | 395396 |
| TTP Identification | Precision | 0.2787 | 0.3065 | 0.3480 | 0.4190 | 0.2387 | 0.2508 | 0.1771 | 0.2111 |
| | Recall | 0.2270 | 0.2163 | 0.1759 | 0.1558 | 0.1846 | 0.1839 | 0.1414 | 0.1358 |
| | Time (s) | 817.62 | 745.63 | 5097.32 | 3582.67 | 2254.21 | 1614.50 | 1889.00 | 1869.12 |
| | Tokens | 466426 | 455265 | 432593 | 432576 | 453282 | 449729 | 493860 | 498774 |
| Content: Threat Actor | Relevance | 1.547 | 2.902 | 3.964 | 4.229 | 3.964 | 4.229 | 2.475 | 4.098 |
| | Accuracy | 1.528 | 2.738 | 3.656 | 3.889 | 3.655 | 3.889 | 2.405 | 3.685 |
| | Comp. | 1.145 | 2.603 | 3.165 | 3.631 | 3.165 | 3.631 | 1.755 | 3.678 |
| | Clarity | 2.019 | 3.776 | 4.753 | 4.814 | 4.752 | 4.814 | 3.699 | 4.678 |
| | Coherence | 1.734 | 3.584 | 4.731 | 4.810 | 4.731 | 4.810 | 3.405 | 4.608 |
| | Attribution | 1.140 | 2.149 | 2.968 | 3.394 | 2.967 | 3.394 | 1.832 | 3.280 |
| Content: Root Cause | Relevance | 3.686 | 3.737 | 4.733 | 4.809 | 4.533 | 4.442 | 3.808 | 3.889 |
| | Accuracy | 3.458 | 3.840 | 4.627 | 4.733 | 4.371 | 4.467 | 3.819 | 3.966 |
| | Comp. | 3.362 | 3.147 | 4.576 | 4.685 | 4.286 | 4.380 | 3.441 | 3.344 |
| | Clarity | 3.932 | 4.000 | 4.818 | 4.897 | 4.629 | 4.731 | 4.163 | 4.229 |
| | Coherence | 3.753 | 3.885 | 4.800 | 4.885 | 4.586 | 4.711 | 3.988 | 4.080 |
| | Attribution | 3.612 | 3.907 | 4.418 | 4.491 | 4.323 | 4.456 | 3.667 | 3.768 |

GPT-4o (FT) refers to Finetuned GPT-4o-2024-08-06; GPT-4o-mini (FT) refers to Finetuned GPT-4o-mini-2024-07-18.
Comp.: Comprehensiveness. Time (s) and Tokens are rounded. Content scores are averages.
IoC Ext.stands for IoC Extraction.

# E   Case Study Examples

## E.1   Metric Comparision

This example is to illustrate why *model-centric* metrics fall short in analysis. The reference article is about threat actor Water Gamayun:

> Researchers at Trend Research uncovered a campaign by the Russian threat actor Water Gamayun, exploiting a zero-day vulnerability in the Microsoft Management Console (MMC), tracked as CVE-2025-26633. The group, also known as EncryptHub and Larva-208, utilized a technique called MSC EvilTwin to manipulate .msc files and the Multilingual User Interface Path (MUIPath) to download and execute a malicious payload, maintain persistence, and steal sensitive data from infected systems. The MSC EvilTwin technique involves creating two .msc files with the same nameone legitimate and one maliciousand using the MUIPath feature to load the malicious file when the legitimate one is executed. The MSC EvilTwin loader, a trojan loader written in PowerShell, was employed to weaponize these techniques. It starts with a digitally-signed MSI file that masquerades as legitimate software, which then fetches the loader from the attacker's server. The loader uses mock trusted directories and manipulates whitespace in file paths to trick applications into loading malicious files from alternate locations. The malicious .msc file, containing the attacker's server address, is loaded when the non-malicious .msc file is executed, leading to the execution of commands such as downloading and executing the Rhadamanthys stealer downloader.Water Gamayun's arsenal includes the EncryptHub stealer, DarkWisp backdoor, SilentPrism backdoor, MSC EvilTwin loader, Stealc, and Rhadamanthys stealer. The campaign is noted for its active development, employing multiple delivery methods and custom payloads designed to maintain persistence and exfiltrate sensitive data to the attackers' servers. Microsoft and Trend Zero Day Initiative (ZDI) disclosed the vulnerability and released a patch on March 11 to mitigate the threat.

The summarized context and results are shown as S-1 and S-2. As shown in Table 7, despite nearidentical BERTScore and slightly lower ROUGE-L, the longer S-2 is favoured by analysts because the S-2 itself provide additional context for deeper analysis for the threat actor *Water Gamayun* and analysts are able to use this context.

## E.2   TRA Case Study

This section illustrates some case study examples and more granular feedback comments from senior human experts, showcasing the advantages of our TRA framework and corresponds the Section 5.2 in our main paper: (1) **TRA can handle cases that contain information ignored by human experts**, such as the case detailed below in Example B.1: *"I was impressed that the model surfaced the fact that 'Many MikroTik devices shipped with a hardcoded admin account with a blank password, which was exploited by the threat actors.' I actually read several articles myself but did not surface this detail."* (2) **TRA can handle tricky issues** like Example B.2.1 and Example B.2.2. And human experts give comments for Example B.2.1: *"The Ivanti article that Caitlin and I just published was a little tricky because it could be Silk Typhoon but it was only medium confidence. The output did a great job of articulating that and providing information about the suspected groups. I was surprised to see they pulled info about the CVE from another 3rd party and not CISA directly (they did pull recs from CISA though).* And Example B.2.2: *It was a little bit more convoluted and pulled "too much" in.. it was a similar situation where there was suspected overlap but unconfirmed. However, the (mitigation) recommendations pulled in for this one were really good!* Based on the feedback commments, TRA can handle tricky issues for threat research, and achieve the expert level.(3) **TRA performs with better automation and more clear**: *I asked my team before this meeting. One of the people said that it is actually more clear than the original articles in some cases. Thats great feedback. It feels easier to me having all of the pieces - IoCs, Detections, Mitigations, MITRE TTPs - in one place instead of having to click around and switch between different tools.*

Table 7: Metrics vs. human assessment for two Water Gamayun summaries.

| Summary | BERT-Score $F_1$ | ROUGE-L $F_1$ | Human Expert verdict |
|---|---|---|---|
| **S1**: Researchers uncovered the attack was carried out by the Russian threat actor Water Gamayun, also known as EncryptHub and Larva-208. The group, exploiting a zero-day vulnerability in Microsoft Management Console (MMC), tracked as CVE-2025-26633, utilised a technique called MSC EvilTwin to manipulate two .msc files with the same name, one legitimate and one malicious, and abused the MUIPath feature to fetch and execute the malicious file, maintain persistence, and exfiltrate sensitive data. | 0.710 | 0.342 | Sparse / Not actionable |
| **S2**: The attack was carried out by the Russian threat actor Water Gamayun, also known as EncryptHub and Larva-208. Water Gamayun is a Russian threat actor known for sophisticated cyber espionage tactics. Recently, they have been linked to the exploitation of a zero-day vulnerability in the Microsoft Management Console (MMC), identified as CVE-2025-26633. This vulnerability allows attackers to manipulate .msc files and the Multilingual User Interface Path (MUIPath) to execute malicious code, enabling data theft and persistence on compromised systems. The group employs advanced techniques, such as using a trojan loader to introduce both clean and malicious .msc files, tricking the MMC into executing the harmful version. This method helps them bypass traditional security measures. The MSC EvilTwin loader, written in PowerShell, facilitates the download and execution of various malware payloads, including EncryptHub stealer and DarkWisp backdoor, which are designed to steal sensitive information and maintain control over infected systems. Water Gamayun's operations are characterized by their ability to blend malicious activities with normal system operations, making detection challenging. They have been active since at least April 2024, continuously refining their methods and leveraging legitimate Windows components to execute their payloads stealthily. Microsoft has released security updates to address this vulnerability, and organizations are urged to apply these updates and implement additional security measures to mitigate the threat. | 0.715 | 0.306 | Preferred / Actionable |

### Criteria for Threat Actor

**Relevance:** Measures how closely the generated context aligns with the key details of the original article (e.g., aliases, motivations, tactics).

- 1: Unrelated or fails to mention key aspects.
- 2: Limited relevance; misses critical details.
- 3: Moderately relevant; covers some aspects but lacks depth.
- 4: Mostly relevant; minor omissions or inaccuracies.
- 5: Highly relevant; fully aligns with the original article.

**Accuracy:** Assesses the factual correctness of the generated context compared to the original article.

- 1: Factually incorrect or inconsistent.
- 2: Contains significant inaccuracies.
- 3: Moderately accurate; minor inconsistencies.
- 4: Mostly accurate; very few errors.
- 5: Completely accurate; perfectly reflects the original article.

**Comprehensiveness:** Evaluates the extent to which the generated context covers all critical details from the original article.

- 1: Highly incomplete; critical details missing.
- 2: Covers only minimal details; significant gaps.
- 3: Moderately comprehensive; some key details missing.
- 4: Comprehensive; minor omissions.
- 5: Fully comprehensive; captures all essential details.

**Clarity:** Measures how clear and understandable the generated context is.

- 1: Poorly written, unclear, and difficult to understand.
- 2: Significant clarity issues; partially understandable.
- 3: Moderately clear; some ambiguities.
- 4: Mostly clear; minor issues.
- 5: Perfectly clear; highly readable and easily understandable.

**Coherence:** Assesses the logical structure and flow of the generated context.

- 1: Disorganized and difficult to follow.
- 2: Significant coherence issues; scattered information.
- 3: Moderately coherent; inconsistent flow.
- 4: Mostly coherent; well-organized with minor issues.
- 5: Fully coherent; logically structured and easy to follow.

**Attribution:** Evaluates whether the generated context properly attributes information to the original article.

- 1: Information is unverified or unattributed.
- 2: Major attribution issues; many details are not clearly linked.
- 3: Moderately attributable; some details lack clear source references.
- 4: Mostly attributable; minor gaps in linking information.
- 5: Fully attributable; all details are clearly linked to the original article.

### Criteria for Root Cause

**Relevance:**

- 1: Unrelated to the root cause of the incident or fails to mention key aspects of the malware's role.
- 2: Limited relevance; partial or tangentially related information but misses key aspects.
- 3: Moderately relevant; covers some important aspects but lacks depth.
- 4: Mostly relevant; covers most key aspects with minor omissions or inaccuracies.
- 5: Highly relevant; fully aligns with the incident's root cause, addressing all key aspects comprehensively.

**Accuracy:**

- 1: Factually incorrect or inconsistent with known details about the malware and the incident's root cause.
- 2: Significant inaccuracies or contradictions, though some elements are correct.
- 3: Moderately accurate; factual but contains minor inconsistencies.
- 4: Mostly accurate; aligns well with known information with very few errors.
- 5: Completely accurate; perfectly reflects the root cause of the incident.

**Comprehensiveness:**

- 1: Highly incomplete; critical details missing.
- 2: Covers only minimal details; significant gaps remain.
- 3: Moderately comprehensive; includes some critical details but misses others.
- 4: Comprehensive; covers most critical details with minor omissions.
- 5: Fully comprehensive; captures all essential details.

**Clarity:**

- 1: Poorly written, unclear, and difficult to understand.
- 2: Significant clarity issues; partially understandable.
- 3: Moderately clear; some ambiguities.
- 4: Mostly clear; minor issues.
- 5: Perfectly clear; highly readable and easily understandable.

**Coherence:**

- 1: Explanation lacks logical flow.
- 2: Some coherence, but there are significant gaps.
- 3: Generally coherent; some weak links.
- 4: Mostly coherent; strong logical connection.
- 5: Fully coherent; clear and logical connection.

**Attribution:**

- 1: Completely incorrect attribution.
- 2: Significant attribution errors; misidentified threat actor.
- 3: Basic attribution; minor inaccuracies.
- 4: Mostly correct attribution.
- 5: Perfect attribution; clearly identifies the threat actor.

## Example B.1: MikroTik Botnet Incident via Misconfigured SPF DNS Records

**Source:** https://www.bleepingcomputer.com/news/security/mikrotik-botnet-uses-misconfigured-spf-dns-records-to-spread-malware

### Related articles (describing the same threat)

- **Original URL of the source:** https://blogs.infoblox.com/threat-intelligence/one-mikro-typo-how-a-simple-dns-misconfiguration-enables-malware-delivery-by-a-russian-botnet

- **Link to source url:** https://securityonline.info/13000-mikrotik-routers-hijacked-for-global-malspam-operation

- **Link to source url:** https://cybersecuritynews.com/botnet-malware-exploit-13000-mikrotik-device

### Enriched Doc (enrichments marked with *content*(link))

**Incident:** MikroTik Botnet via Misconfigured SPF DNS Records
**Root Cause:** The root cause of the incident is the misconfiguration of SPF (Sender Policy Framework) DNS records. Specifically, the SPF records for about 20,000 domains were configured with the overly permissive '+all' option, which allows any server to send emails on behalf of those domains. This misconfiguration enabled the threat actors to spoof emails and bypass email protections.
**Threat actor/group/campaign:** The threat actor is linked to a botnet operation involving approximately 13,000 hijacked MikroTik devices. The command and control (C2) server used in the attack is tied to Russian hackers. The botnet also configured the devices as SOCKS4 proxies to launch DDoS attacks, send phishing emails, exfiltrate data, and mask the origin of malicious traffic. The C2 server IP is 62.133.60.137 hosted by Global Connectivity Solutions. The emails impersonated DHL and contained a zip file with an obfuscated JavaScript file that executed a PowerShell script to connect to the C2 server. The campaign also exploited critical vulnerabilities in MikroTik routers, stemming from outdated firmware or misconfigured security settings. The botnet comprises MikroTik routers with various firmware versions, including recent ones, and exploits a remote code execution vulnerability. Infoblox Threat Intel highlighted the use of trojan malware in the campaign. Many MikroTik devices shipped with a hardcoded admin account with a blank password, which was exploited by the threat actors.
**Organization/industry/location:** Not specified
**Start date End date:** The malspam campaign was active in late November 2024.
More information on next page...

---

**Example B.1: MikroTik Botnet Incident via Misconfigured SPF DNS Records**

**MITRE TTPs:**

- T1071.001: Application Layer Protocol: Web Protocols; Confidence: High. Justification: The report mentions the use of PowerShell scripts to establish a connection to the threat actors command and control (C2) server.

- T1566.002: Phishing: Spearphishing Link; Confidence: Medium. Justification: The report describes the use of malspam emails impersonating DHL Express to deliver fake freight invoices with a malicious payload.

- T1090.002: Proxy: External Proxy; Confidence: Medium. Justification: The botnet configured MikroTik devices as SOCKS4 proxies to mask the origin of malicious traffic.

- T1072: Software Deployment Tools; Confidence: Medium. Justification: The use of PowerShell scripts to deploy malware indicates the use of software deployment tools.

- T1090.001: Proxy: Internal Proxy; Confidence: Medium. Justification: The botnet's configuration as SOCKS4 proxies allowed tens or even hundreds of thousands of compromised machines to use them for network access.

- T1059.007: Command and Scripting Interpreter: JavaScript; Confidence: High. Justification: The report mentions the use of an obfuscated JavaScript file to execute a PowerShell script.

- T1078: Valid Accounts; Confidence: High. Justification: The threat actors used stolen credentials to access and exploit MikroTik devices.

- T1204.002: User Execution: Malicious File; Confidence: High. Justification: The report mentions the use of zip file attachments containing obfuscated JavaScript files.

**Impact:** The attackers have created a botnet of 13,000 MikroTik devices and are conducting spoofing of roughly 20,000 web domains.

**Mitigation Steps:**
Based on OSINT recommendation dictionary (Guidance for Botnets), the recommendations are:

- Restrict automatic prompts for non-user-initiated file downloads.

- Enable Safe Links protection for links in email messages.

- Enable Safe Attachments in block mode.

- Enable Zero-hour auto purge (ZAP) in Office 365 to quarantine sent mail in response to newly-acquired threat intelligence and retroactively neutralize malicious phishing, spam, or malware messages that have already been delivered to mailboxes.

- Run endpoint detection and response (EDR) in block mode so that Microsoft Defender for Endpoint can block malicious artifacts, even when your non-Microsoft antivirus does not detect the threat or when Microsoft Defender Antivirus is running in passive mode.

**IoCs:**

- IP: 62.133.60.137, Publish date: 2025-01-14 [In this link], Verified via VirusTotal.

**Example B.2.1: Ivanti Connect Secure Zero-Day Exploitation**

**Source:** https://www.bleepingcomputer.com/news/security/ivanti-warns-of-new-connect-secure-flaw-used-in-zero-day-attacks

## Related articles (describing the same threat)

- https://www.cisa.gov/cisa-mitigation-instructions-cve-2025-0282
- https://www.bleepingcomputer.com/news/security/google-chinese-hackers-likely-behind-ivanti-vpn-zero-day-attacks [link to source url]
- https://www.rapid7.com/blog/post/2025/01/08/etr-cve-2025-0282-ivanti-connect-secure-zero-day-exploited-in-the-wild [link to source url]
- https://www.tenable.com/blog/cve-2025-0282-ivanti-connect-secure-zero-day-vulnerability-exploited-in-the-wild [link to source url]
- https://www.wiz.io/blog/cve-2025-0282-and-cve-2025-0283-critical-ivanti-0days-exploited-in-the-wild [link to source url]
- https://cloud.google.com/blog/topics/threat-intelligence/ivanti-connect-secure-vpn-zero-day
- https://thehackernews.com/2025/01/ivanti-flaw-cve-2025-0282-actively.html [link to source url]
- https://www.ivanti.com/blog/security-update-ivanti-connect-secure-policy-secure-and-neurons-for-zta-gateways [Original link with source]

## Enriched Doc (enrichments marked with *content*(link))

**Incident:** Ivanti Connect Secure Zero-Day Exploitation
**Root Cause:** The root cause of the incident is a critical stack-based buffer overflow vulnerability (CVE-2025-0282) in Ivanti Connect Secure before version 22.7R2.5, Ivanti Policy Secure before version 22.7R1.2, and Ivanti Neurons for ZTA gateways before version 22.7R2.3. This vulnerability allows an unauthenticated attacker to remotely execute code on affected devices. Additionally, CVE-2025-0283, a privilege escalation vulnerability, affects Ivanti Connect Secure appliances and is critical to patch as it might be exploited in conjunction with CVE-2025-0282 for more complex attack chains.
**Threat actor/group/campaign:** The specific threat actor or group behind the exploitation of this vulnerability is not mentioned in the report. However, Ivanti is working with Mandiant, the Microsoft Threat Intelligence Center, external security partners, and law enforcement agencies to investigate the attacks. Mandiant suspects the involvement of China-linked espionage groups UNC5337 and UNC5221.
**Organization/industry/location:** The victims are organizations using Ivanti Connect Secure appliances. The specific organizations or industries affected are not disclosed in the report. However, Ivanti mentioned a limited number of customers were exploited.
**Start date End date:** The exact start date of the attacks is not mentioned, but the report was published on January 8, 2025, indicating that the attacks were detected and disclosed around this time. Mandiant observed attackers leveraging the vulnerability since mid-December.

---

**Example B.2.1: Ivanti Connect Secure Zero-Day Exploitation**

**MITRE TTPs:**

- T1203: Exploitation for Client Execution; Confidence: High. Justification: The report mentions that the vulnerability allows unauthenticated attackers to remotely execute code on devices, which aligns with this TTP.

- T1078: Valid Accounts; Confidence: Medium. Justification: The report mentions a second vulnerability (CVE-2025-0283) that allows an authenticated local attacker to escalate their privileges, which aligns with this TTP.

- T1071: Application Layer Protocol; Confidence: Medium. Justification: The report mentions that the attackers used the vulnerability to install malware on appliances, which aligns with this TTP.

- T1070: Indicator Removal on Host; Confidence: High. Justification: The attackers modified iptables rules and remounted the drive as 'read-write' to allow malware deployment, which aligns with this TTP.

- T1059: Command and Scripting Interpreter; Confidence: High. Justification: The attackers deployed a web shell to compromised components, which aligns with this TTP.

- T1003: Credential Dumping; Confidence: High. Justification: The attackers used the Dryhook malware to capture usernames and passwords during standard authentication processes, which aligns with this TTP.

**Impact:** The exact number of records or devices impacted is not specified in the report. However, the exploitation of the vulnerability allowed attackers to install malware on affected appliances. The attackers installed custom malware called 'Dryhook' and 'Phasejam'. Additionally, the SPAWN ecosystem of malware, including SPAWNANT, SPAWNMOLE, and PAWNSNAIL, was identified on compromised systems.

**Mitigation Steps:**
Based on OSINT recommendation dictionary (Mitigate zero-day vulnerabilities), the recommendations are:

- Use Microsoft Defender Vulnerability Management to identify and address zero-day vulnerabilities.

**IoCs:**

- **hash_md5:** e7d24813535f74187db31d4114f607a1, Publish date: 2025-01-08 [In this link], Verified via VT

- **hash_md5:** a638fd203ddb540d0484d8e00490df06, Publish date: 2025-01-08 [In this link], not included in VT database

- **hash_md5:** d18e5425ecd9608ecb992606b974e15d, Publish date: 2025-01-08 [In this link], not included in VT database

- **hash_md5:** 61bb586dc4e047ab081ef6ca65684e48, Publish date: 2025-01-08 [In this link], not included in VT database

## Example B.2.2: EAGERBEE Malware: Updated Arsenal for Attacking ISPs & Government Entities

**Source:** https://gbhackers.com/eagerbee-malware

## Related Articles (Describing the Same Threat)

- https://www.bleepingcomputer.com/news/security/eagerbee-backdoor-deployed-against-middle-eastern-govt-orgs-isps [link to source url]
- https://fieldeffect.com/blog/eagerbee-isps-government-entities [link to source url]
- https://www.darkreading.com/cyberattacks-data-breaches/eagerbee-backdoor-middle-east-isps-government-targets [link to source url]
- https://socprime.com/blog/eagerbee-malware-detection
- https://securelist.com/eagerbee-backdoor/115175 [Original Link with Source]
- https://thehackernews.com/2025/01/new-eagerbee-variant-targets-isps-and.html [link to source url]

## Enriched Doc (Enrichments Marked with *content*(link))

**Incident:** EAGERBEE Malware Updated Its Arsenal to Attack ISPs & Government Entities

**Root Cause:** The root cause of the incident is the exploitation of the ProxyLogon vulnerability in Exchange servers (CVE-2021-26855), which allowed attackers to deploy the EAGERBEE backdoor. Additionally, the attackers used service manipulation, DLL hijacking techniques, and privilege escalation techniques to load malicious DLLs and maintain persistence. The attackers deployed a service injector ('tsvipsrv.dll') and the 'ntusers0.dat' payload, which leveraged the 'SessionEnv' service to execute. The backdoor appears on the infected system as 'dllloader1x64.dll' and uses 'ssss.dll' to manage plugins. Once deployed, the backdoor collects extensive system information, such as the local computers NetBIOS name, OS details, and network addresses. It establishes communication with a command-and-control (C2) server, leveraging encrypted protocols like SSL and TLS encryption.

- Additional context for ProxyLogon, EAGERBEE:

ProxyLogon refers to a critical vulnerability (CVE-2021-26855) in Microsoft Exchange Server, which allows attackers to perform server-side request forgery (SSRF) attacks. This vulnerability has been leveraged by multiple threat actors, including state-sponsored groups and financially motivated cybercriminals, to gain unauthorized access to vulnerable Exchange servers, leading to further exploitation and malware deployment.EAGERBEE is a sophisticated backdoor malware discovered targeting ISPs and governmental entities in the Middle East. It uses a unique service injector to embed itself into running services, employing a DLL file ("tsvipsrv.dll") and a payload file ("ntusers0.dat") for installation via the SessionEnv service. Once installed, it collects system information and establishes a connection with its C2 server, often using SSL/TLS encryption. EAGERBEE continuously operates, retrieving configurations, and using the Plugin Orchestrator module to manage additional plugins for malicious activities, such as file system manipulation, process management, and remote access. The malware injects malicious code into legitimate processes to evade detection and shares similarities with the CoughingDown threat group. Some instances of EAGERBEE were deployed through the ProxyLogon vulnerability, highlighting the necessity of patching known exploits. Researchers have observed the use of legitimate Windows services for loading malicious DLLs, complicating attribution efforts.

**Example B.2.2: EAGERBEE Malware: Updated Arsenal for Attacking ISPs & Government Entities**

**Threat Actor/Group/Campaign:** The attack is linked to the CoughingDown threat group, also known as TA428, as suggested by consistent service creation and C2 domain overlap. The CoughingDown Core Module was also identified. The use of the most recent malware iteration is attributed with medium confidence to a hacking group tracked as CoughingDown. EAGERBEE was initially identified by Elastic Security Labs, linked to a state-sponsored cyber-espionage group known as REF5961. It was observed in cyber-espionage attacks against Southeast Asian government agencies and linked to the Chinese nation-backed hacking collective, which Sophos tracked as Crimson Palace. Previous researchers had attributed EAGERBEE to Chinese threat group Iron Tiger (APT27), one of numerous groups that often collaborate with other China-backed state-sponsored actors. Western ISPs and telecommunications service providers (TSP) were heavily targeted by PRC-backed groups, including Salt Typhoon, who potentially obtained metadata associated with communication habits of millions of the compromised providers customers . The new variant of EAGERBEE (aka Thumtais) was observed in attacks by a Chinese state-aligned threat cluster tracked as Cluster Alpha , which overlaps with groups like BackdoorDiplomacy, REF5961, Worok, and TA428. BackdoorDiplomacy exhibits tactical similarities with another Chinese-speaking group codenamed CloudComputating (aka Faking Dragon), attributed to a multi-plugin malware framework referred to as QSC.

**Organization/Industry/Location:** The targeted victims are ISPs and government entities in the Middle East and East Asia.

**Start Date - End Date:** The exact start and end dates of the attack are not specified in the report.

**MITRE TTPs:**

- T1059: Command and Scripting Interpreter; Confidence: Medium; Justification: The use of a service injector and manipulation of services to load malicious DLLs suggests the use of scripting or command execution.

**Impact:** The exact number of records leaked or financial losses is not specified in the report.

**Mitigation Steps:** Based on MDTI profile, we recommends the following mitigations to reduce the impact of this threat:

- Keep public-facing servers up to date to defend against malicious activity.
- Utilize a vulnerability management system such as Microsoft Defender Vulnerability Management.
- Turn on cloud-delivered protection in Microsoft Defender Antivirus.
- Enable real-time protection in Microsoft Defender Antivirus.
- Implement anomaly detection policies in Microsoft Defender for Cloud Apps.

**IoCs:**

- ip: 45.90.58.103, Publish date: 2025-01-07 [In this link], Verified via VT
- ip: 185.82.217.164, Publish date: 2025-01-07 [In this link], Verified via VT
- ip: 62.233.57.94, Publish date: 2025-01-07 [In this link], Verified via VT
- ip: 82.118.21.230, Publish date: 2025-01-07 [In this link], Verified via VT
- domain: www.socialentertainments.store, Publish date: 2025-01-06 [In this link], Verified via VT

