# OpenReview forum: "CyberThreat-Eval: Can Large Language Models Automate Real-World Threat Research?"
_TMLR — Accepted by TMLR_

### Review · Reviewer_HcGN · 2025-09-22

**Summary Of Contributions:**

Cyber Threat Intelligence (CTI) plays a crucial role in identifying and analysing emerging cybersecurity threats. Analysts performing CTI are often required to process large amounts of data in order to make comprehensive and well-informed reports. With the emergence of Large Language Models (LLMs), the automation of CTI tasks no longer seems infeasible; however, we need well-designed, reliable, and comprehensive benchmarks to evaluate the performance of such systems. There are several existing benchmarks designed for this task, but as the authors point out, they either 1) have unrealistic task formats or 2) propose model-centric, not analyst-centric, performance measures. The authors also point out that (3) current benchmarks do not cover the end-to-end CTI workflow, which they define as three consecutive stages: *triage*, *deep research* and *Threat Intelligence (TI) drafting*.
To overcome the above three limitations, the authors propose the CyberThreat-Eval benchmark, a novel framework for evaluating LLMs on realistic (1), expert-annotated (2) end-to-end CTI reporting tasks, which cover all three stages (3).
The authors proceed to demonstrate the effectiveness of their benchmark by evaluating different variations of the GPT-4o and o3 families of models. Their results consistently show limitations of current LLMs, which they further demonstrate through detailed case studies in the appendix.

Finally, they describe the high-level architecture of an LLM-based threat research framework, called Threat Research Agent (TRA), which outperforms vanilla LLMs across practically all metrics on the CyberThreat-Eval benchmark.

**Additional Comments:**

None

**Audience:**

Yes

**Audience Explanation:**

Automation of CTI tasks is a relevant application of LLMs and has the potential to enhance the cyber resilience of many industry actors. However, as the authors rightly point out, this problem is not yet solved. CyberThreat-Eval offers a reliable, realistic and objective metric to evaluate existing and new solutions to this problem.

**Claims And Evidence:**

Yes

**Claims Explanation:**

Claims about the limitations of current benchmarks are well-founded and supported by tables and figures. To my knowledge, they did not leave any existing work out.
Claims about the CyberThreat-Eval benchmark are well-demonstrated, albeit their evaluation criteria are not always sufficiently described (see W1 below).
Claims about the TRA are also supported by evidence, and the authors clearly demonstrate extensive testing in this regard

**Requested Changes:**

W1: In Section 4.1, the authors mention using the LLM-as-Judge paradigm to evaluate the final outputs of the different language models in their experiments.
I believe it would benefit the paper if the authors included more details on the exact setup of this evaluation method and what actions they took to minimise the likelihood of incorrect judgments. I would also be interested to know to what extent the generated judgements agree with human expert judgements.
W2: The "Data Size" column in Table 3 is difficult to comprehend. Please name the metrics in each row (eg, Triage: 488 *articles*)
W3: To me, it was somewhat unclear whether in the Deep Search phase, the task is to evaluate a *URL* or the *article* to which the URL points. Similarly, I find it difficult to understand what "Avg. URLs processed (w. input)" means in Table 4.
My intuition tells me that the evaluated model first needs to decide whether a given URL should be investigated (URLs processed metric) and then, whether the document behind the URL contains relevant new information (URLs w. additional info metric). However, I do not find a clear description of this in the article. I believe Section 3.3.2 could be improved in this regard.

In my opinion, all of these modifications would be beneficial to the article, but I leave it to the author's judgment to implement them. I will recommend this article for acceptance as I believe it presents a well-established and novel evaluation benchmark for an important and impactful branch of LLM research.

---

> ### Author Response · Authors · 2025-09-27
>
> We would like to express our sincere gratitude for your time and for providing such insightful and constructive feedback on our manuscript.  Below, we detail the specific revisions made in response to each of your points (W1, W2, and W3).
> ## W1
> Thank you for this important point. To enhance the reliability of the judgments made by LLMs, we provided the judge LLM with a clear, rule-based rubric drafted by our security experts. For each score, we have a clear and detailed rubric. In addition, the output of the model should contain justification part to enhance the reliablity of our method. We tested it on approximately 100 artile examples from human experts for 2 rounds. Our human experts then reviewed these LLM-generated judgments and provided detailed feedback on cases where the model's reasoning deviated from their own idea. Based on this expert feedback of the two rounds, we adjusted and clarified the evaluation criteria in our method. We improved the alignment rate between human experts and LLM from around 80% to over 95% after the two-round refinement.
> ## W2
> We thank the reviewer for your suggestion to improve the clarity of our data description. We agree that adding explicit units is essential. Accordingly, we have revised corresponding part of Table 3.
>
> The revised Table 3 now appears as follows:
> | Category | Task / Component | Data Size |
> | :-------------- | :------------------- | :------------ |
> | Triage          | Priority Scores      | 488 articles  |
> | Deep Search     | Input URLs           | 55 URLs       |
> | TI Drafting     | IoCs                 | 1310 IoCs     |
> | TI Drafting     | TTPs Identification                 | 1565 TTPs     |
> ## W3
> Thank you for highlighting the ambiguity in our description of the Deep Search task.
> **Clarification of the Evaluation Target**: We state that the evaluation target is the content of the article to which a URL points, not the URL string itself. Here, the URLs are seen as input. For example, when given an input, the users should extract the article content behind the URLs for finding more relevant URLs.
> The metric "Avg. URLs processed (w. input)" is defined as the average number of unique URLs (the initial input URL plus all discovered candidate URLs) whose content is fetched and analyzed by the LLM. For each "processed" URL, its content is then evaluated to determine if it contains additive information, which in turn yields the "Avg. URLs w. additional info" metric. This metric is used to evalute how many URLs are analyzed in an incident.
>
> Thank you again for your valuable feedback and your support for our work. We hope that the revisions have made our manuscript more solid and clearer.

---

### Review · Reviewer_Dz9a · 2025-09-26

**Summary Of Contributions:**

Contribution and strengths:

1. The paper addresses a current gap by focusing on real-world analyst workflows rather than artificial benchmark tasks. The three-stage workflow appears well-motivated and grounded in actual practice.

2. Using actual workflow data from a world-leading technology company provides authenticity that synthetic benchmarks lack. The expert annotations add credibility.

Weaknesses:

1. The entire benchmark is derived from one company's workflow, different organizations likely have substantially different CTI processes.

2. Another concern is that the sample size is quite small, this may limit robust evaluation, and limit the applicability.

3. Human expert annotations are treated as gold standard, but inter-annotator agreement isn't reported. CTI analysis involves significant subjectivity, which may also limit the faithfulness of this benchmark.

**Audience:**

Yes

**Audience Explanation:**

The benchmark address some serious gaps in the field, to my best understanding.

**Claims And Evidence:**

Yes

**Claims Explanation:**

Yes, the expriments and description support the claims.

**Requested Changes:**

I hope the authors could address my concerns as following:

1. Would the sample size limit statistical power and generalizability?

2. Would the individual evaluation not be robust and not sufficient for the benchmark?

3. Would LLM-as-judge for content evaluation introduce  bias and inconsistency?

---

> ### Author Response · Authors · 2025-10-04
> **Official Comment by Authors on Weaknesses by Reviewer Dz9a**
>
> ## W1: About end-to-end workflow
> We thank the reviewer for the comment regarding generalizability. We agree that CTI workflows can vary across organizations. In fact, a key challenge in this domain is the lack of a standardized, academically defined end-to-end paradigm for the threat research process.
> Much of the existing research focuses on evaluating discrete subtasks (like IoC extraction or summarization) in isolation, without situating them within a cohesive, operational workflow. Our paper is to address this ambiguity by proposing a concrete, structured, and practical paradigm for the end-to-end CTI workflow. This paradigm is not arbitrary; it is grounded in the mature, real-world practices of a large-scale, industry-leading cybersecurity operation, providing an authentic and replicable model for what a comprehensive threat analysis entails.
> By first defining this operational paradigm, we were then able to construct the CyberThreat-Eval benchmark to rigorously evaluate LLM capabilities against these realistic stages. Therefore, while the specific tools or priorities might differ between organizations, our benchmark provides a foundational and transparent framework that the research community can now test, adapt, critique, and build upon. We believe establishing such a concrete, industrially-validated paradigm is a necessary first step towards more standardized and generalizable CTI evaluations.
>
> ## W2: Question about sample size
> We thank the reviewer for raising this important question regarding the sample size and its potential impact on statistical power and generalizability. Here we coment on weakness 2 as well as answer the question: "Would the sample size limit statistical power and generalizability?"
> We acknowledge that the dataset size is modest when compared to benchmarks for general LLM tasks. However, this must be contextualized within the specific nature of the Cyber Threat Intelligence (CTI) or cybersecurity-related domain, where the complexity and cost of expert annotation are significantly higher. The tasks within CyberThreat-Eval, particularly in-depth content generation and multi-faceted priority scoring, demand a profound level of domain expertise and analysis, making large-scale annotation challenging.
> In addition, the overall data sample size is much smaller than general domains (e.g., there are summarized 862 threat actors and ~3.6k malwares in the database Malpedia [1];)
> For context, our dataset size is comparable to, and in some cases larger than, other highly-regarded, expert-annotated CTI benchmarks that also focus on complex reasoning tasks. For instance, CTIBench [2] is built upon 2,500 instances for its CTI QA tasks, and 1k instances for malwares, 397 questions for unique attack techniques; and CTISum [3] uses 1,345 reports for its summarization evaluation.
>
> Therefore, considering the specialized nature of CTI tasks and referencing established prior work, we argue that the size of our benchmark dataset is sufficient for the end-to-end CTI-based task.
>
> ## W3: Question about inter-annotator agreement of human experts
> We thank the reviewer for this point regarding annotation reliability. For the robustness of our ground truth, we employed a structured, multi-stage consensus-building process.
> To ensure the reliability of the dataset, labels were applied by an analyst with several years of professional experience in cybersecurity-related open-source threat intelligence. In cases where the classification was straightforward, the analyst independently assigned the appropriate label. When ambiguities arose, the case was escalated to a broader group discussion within the team. These discussions allowed multiple analysts to review the evidence, share perspectives, and reach a consensus on the most appropriate labeling. This process provided both consistency across the dataset and an internal mechanism for quality control, helping to mitigate individual bias and strengthen confidence in the final labels.
>
> References:
>
> [1] Malpedia. https://malpedia.caad.fkie.fraunhofer.de/
>
> [2] Md Tanvirul Alam, Dipkamal Bhusal, Le Nguyen, Nidhi Rastogi. CTIBench: A Benchmark for Evaluating LLMs in Cyber Threat Intelligence. NeurIPS. https://arxiv.org/abs/2406.07599.
>
> [3] Wei Peng, Junmei Ding, Wei Wang, Lei Cui, Wei Cai, Zhiyu Hao, Xiaochun Yun. CTISum: A New Benchmark Dataset For Cyber Threat Intelligence Summarization. https://arxiv.org/abs/2408.06576.

---

> > ### Author Response · Authors · 2025-10-04
> > **Additional Comment on Requested Changes**
> >
> > We thank the reviewer Dz9a for the constructive comments on our paper. For the requested changes part, we detail the specific comments in response to each of your points below.
> >
> > For requested changes "1.Would the sample size limit statistical power and generalizability?" and "2. Would the individual evaluation not be robust and not sufficient for the benchmark?", the answers are given in W2 and W3 respectively.
> >
> > For "Would LLM-as-judge for content evaluation introduce bias and inconsistency?", we thank the reviewer for this question regarding the LLM-as-Judge paradigm in our paper. To enhance the reliability of the judgments made by LLMs, our method on this consisted of the following key stages:
> >
> > 1. Initial Rubric Development: We began by drafting a clear, rule-based rubric and criteria in collaboration with our security experts. A crucial component of this was the requirement for the judge LLM to provide a justification for its decisions. And each score of the rubric should be aligned with the standard of the experts.
> >
> > 2. We then tested this initial setup on a set of ~100 diverse article examples for 2 rounds respectively. Our human experts subsequently reviewed these LLM-generated judgments, providing detailed qualitative feedback on all cases where the LLM's answer deviated from their own expert assessment.
> >
> > 3. Based on the expert feedback from this first round, we systematically adjusted and clarified the evaluation criteria and examples within our justification.
> >
> > This two-round refinement process was highly effective: we demonstrably improved the alignment rate between the LLM-as-Judge and our human experts. The alignment score is given positive if LLM-based score on one article is aligned with the experts. And this score increases from an initial ~80% to a final, stable rate exceeding 95%, which demonstrated its effectiveness.

---

### Review · Reviewer_LWxd · 2025-09-28

**Summary Of Contributions:**

This work introduces a benchmark for cyber threat intelligence that mirrors a real analyst workflow as well as an agentic, human-in-the-loop system (TRA) to execute it. Across commercial and fine-tuned LLMs, they find high recall but noisy precision in triage, consistently weak TTP/IoC extraction, and modest gains in explanatory reporting.
TRA improves precision via external tools and expert checks but at higher latency/token cost.

### Strength
1. Automating CTI workflows represents a high-impact and timely application area for LLM-based agents.
2. The study evaluates the entire CTI pipeline rather than isolating individual components.
3. Beyond proposing a benchmark, the work implements TRA as a real-world instantiation of agentic methods grounded in the benchmark.

### Weaknesses
1. The LLM-as-judge framework lacks reported checks for reliability. In particular, measuring the inter-judge agreement would improve the evaluation setup significantly. Without this, the observed gains may reflect judge-specific biases rather than genuine task improvements.
2. The evaluation metrics do not report the variance across generation seeds, making it hard to judge the robustness of the reported numbers.
3. The fine-tuning setup is under-specified: the paper does not detail tuning objectives, data curation and volume. The mixed results of fine-tuned models are very interesting but hard to interpret without additional details on how fine-tuning was performed.
4. The live web-search component is inherently sensitive to web content drift, but the paper does not describe snapshotting, versioning, or archival strategies to stabilize evaluation once the benchmark is public. As a result, the long-term comparability and fairness of the evaluation setup remain unclear.

**Audience:**

Yes

**Audience Explanation:**

Applying LLM agents to CTI problems is of interest to the TMLR audience.

**Claims And Evidence:**

Yes

**Claims Explanation:**

While there are some weaknesses, I do think the overall results are accurate.

**Requested Changes:**

Reporting inter-judge agreement for experiments evaluating the LLM-as-judge would be a critical addition to the paper. Other additions that would improve the work are:
1. Reported variance across generation seeds.
2. Additional details on how fine-tuning was done.
3. A discussion on how the web-search evaluation will be robust in a public version of the benchmark.

---

> ### Author Response · Authors · 2025-10-07
> **Official Comment by Authors (1/2)**
>
> We thank the reviewer LWxd for the insightful comments and feedbacks on our paper. For the weaknesses and requested changes part, we detail the specific comments in response to each of your points below.
>
> ## W1: LLM-as-judger
> We thank the reviewer for this question regarding the LLM-as-Judge paradigm in our paper. To enhance the reliability of the judgments made by LLMs, our method on this consisted of the following key stages:
>
> - Initial Rubric Development: We began by drafting a clear, rule-based rubric and criteria in collaboration with our security experts. A crucial component of this was the requirement for the judge LLM to provide a justification for its decisions. And each score of the rubric should be aligned with the standard of the experts.
> - Human in the loop testing: We then tested this initial setup on a set of ~100 diverse article examples for 2 rounds respectively. Our human experts subsequently reviewed these LLM-generated judgments, providing detailed qualitative feedback on all cases where the LLM's answer deviated from their own expert assessment. Based on the expert feedback from this first round, we systematically adjusted and clarified the evaluation criteria and examples within our justification.
>
> This two-round refinement process was highly effective: we demonstrably improved the alignment rate between the LLM-as-Judge and our human experts. The alignment score is given positive if LLM-based score on one article is aligned with the experts. And this score increases from an initial ~80% to a final, stable rate exceeding 95%, which demonstrated its effectiveness.
>
> ## Additional details on generation seeds
> We thank the reviewer for the value of reporting variance. To ensure our results are as stable and reproducible as possible, we employed all-the-same parameters for all our generation tasks. Specifically, we set a fixed seed=42 and a very low temperature=0.01 for all LLM calls. This configuration minimizes the stochasticity inherent in the generation process, making our single-run results highly consistent and minimizing variance. Based on your feedback, we will explicitly stated these parameters in our Experiments section to enhance the reproducibility of our work.
>
> Furthermore, it is important to contextualize this point within the scope of our contribution. The primary goal of this paper is not to claim a new state-of-the-art performance by a narrow, incremental margin on previous tasks. Rather, our main contribution is the introduction and validation of a novel benchmark paradigm on CTI domain, and the proposal of TRA, a potential architectural solution that demonstrates a principled way to improve upon baseline LLMs. The performance gains demonstrated by TRA, particularly in complex tasks are substantial and qualitative, not marginal. The large delta between TRA and the baseline models provides strong evidence of the architectural benefits of our approach. This significant improvement is robust against the minor variations that might arise from different generation seeds, making a single, carefully controlled run highly indicative of the framework's effectiveness.
>
> While we believe these settings ensure a high degree of stability, we acknowledge that due to the significant computational costs associated with running extensive experiments on large commercial LLMs, we did not perform multiple runs across all different random seeds to report variance between all the random seeds.
>
> ## The web-search component
> We thank the reviewer for this foresight regarding the long-term stability and reproducibility of live web-search component. We considered this issue mostly on the Deep Search task.
> To address the issue of web content drift and ensure the long-term comparability of CyberThreat-Eval, we have adopted a data snapshotting strategy in this section. Rather than conserving the URLs only, we archived the full HTML content of the web pages that needed during our experiments. This ensures that all future research using our benchmark will be performed against the exact same data as we implemented, guaranteeing fair and reproducible comparisons.

---

> > ### Author Response · Authors · 2025-10-07
> > **Official Comment by Authors (2/2)**
> >
> > This part follows the previous response:
> >
> > ## Additional details on fine-tuning
> > To address this, we include the following details for our fine-tuning process, which was applied to both the GPT-4o and GPT-4o-mini models.
> >
> > Finetuning Objectives: The primary objective was to enhance the LLMs' overall domain-specific capabilities across a diverse range of CTI tasks, rather than optimizing for any single benchmark subtask, so as to generate more robust CTI-related reports.
> >
> > Methods: We employed a standard Supervised Fine-Tuning (SFT) approach. The aggregated dataset was partitioned into training (79%), validation (1%), and testing (20%) sets to monitor the fine-tuning process. In addition, the data format for fine-tuning is OpenAI format.
> >
> > Data Curation and Composition: The fine-tuning dataset was a curated aggregation and collection of multiple proprietary, high-quality datasets used in our internal security operations This multi-task dataset was formatted in the standard OpenAI format, and included a variety of CTI-related tasks, including but not limited to:
> > Question-Answering datasets: like cybersecurity_qa and mitre_questions; CTI Mapping; Natural-Language-to-Query: Datasets for translating natural language questions into Kusto Query Language (KQL) (nl2kql); Function Calling: A dataset for security-related tool use and function calling (security_copilot_function_calling);

---

### Author Response · Authors · 2025-12-18
**Thank you for the Acceptance Decision and Camera-Ready Updates**

We would like to express our sincere gratitude to the AE 6RmH and all three reviewers (LWxd, Dz9a, HcGN) for their time, effort, and invaluable feedback throughout this review process. In preparation for the camera-ready version, we have finalized the revisions based on the critical discussions during the rebuttal phase. Specifically, we have incorporated a detailed introduction and validation of our LLM-as-Judge methodology to demonstrate its reliability. We also added a comprehensive description of the datasets and specific methods used for SFT to clarify our experimental setup in evaluating fine-tuned LLMs. Additionally, we have finalized the preparation of our code repo for open-source, pending final internal code review, to support the reproducibility of our paper. We are excited to share these contribution and findings with the community.

Best, Authors

---

### Decision · Action_Editor_6RmH · 2025-11-20

**Recommendation:** Accept as is

**Audience:**

Yes

**Audience Explanation:**

This study presents an interesting topic that evaluates LLM in the real-world cyberThreat scenario, which draws the attention about the limitation of single LLMs. AE thinks that this topic will attract the groups who are interested in LLMs, and will extend the scope of LLM applications.

**Claims And Evidence:**

Yes

**Claims Explanation:**

This submission proposes a benchmark for cyber threat intelligence, and the evaluation using this benchmark reveals important insights into the limitations of current LLMs. The reviewers raised the concerns about the judge biases, result variance, sample size, golden standard and unclear details etc.. The authors provided the comprehensive feedback and improve the manuscript. AE has checked the details and confirmed the reviewers' final recommendation, supporting the claim with the proper evidence.

---

> ### Author Response · Authors · 2025-12-18
> **Thank You for the Acceptance and Feedback**
>
> We would like to express our sincere gratitude to the AE 6RmH and all three reviewers (LWxd, Dz9a, HcGN) for their time, effort, and invaluable feedback throughout this review process. In preparation for the camera-ready version, we have finalized the revisions based on the critical discussions during the rebuttal phase. Specifically, we have incorporated a detailed introduction and validation of our LLM-as-Judge methodology to demonstrate its reliability. We also added a comprehensive description of the datasets and specific methods used for SFT to clarify our experimental setup in evaluating fine-tuned LLMs. Additionally, we have finalized the preparation of our code repo for open-source, pending final internal code review, to support the reproducibility of our paper. We are excited to share these contribution and findings with the community.
>
> Best,
> Authors